# Disentanglement of Correlated Factors via Hausdorff Factorized Support

**Karsten Roth**[1,][*] **Mark Ibrahim**[2], **Zeynep Akata**[1,3], **Pascal Vincent**[2,4,†], **Diane Bouchacourt**[2,†]
[1]University of Tübingen, ELLIS [2]Meta AI, FAIR [3]MPI for Intelligent Systems
[4]MILA, Université de Montréal/DIRO, CIFAR
Primary contact: `karsten.roth@uni-tuebingen.de`

## Abstract

A grand goal in deep learning research is to learn representations capable of generalizing across distribution shifts. Disentanglement is one promising direction aimed at aligning a model's representation with the underlying factors generating the data (e.g. color or background). Existing disentanglement methods, however, rely on an often unrealistic assumption: that factors are statistically independent. In reality, factors (like object color and shape) are correlated. To address this limitation, we consider the use of a relaxed disentanglement criterion – the Hausdorff Factorized Support (HFS) criterion – that encourages only pairwise factorized *support*, rather than a factorial distribution, by minimizing a Hausdorff distance. This allows for arbitrary distributions of the factors over their support, including correlations between them. We show that the use of HFS consistently facilitates disentanglement and recovery of ground-truth factors across a variety of correlation settings and benchmarks, even under severe training correlations and correlation shifts, with in parts over $+60\%$ in relative improvement over existing disentanglement methods. In addition, we find that leveraging HFS for representation learning can even facilitate transfer to downstream tasks such as classification under distribution shifts. We hope our original approach and positive empirical results inspire further progress on the open problem of robust generalization. Code available at https://github.com/facebookresearch/disentangling-correlated-factors.

## 1 Introduction

Disentangled representation learning (Bengio et al., 2013; Higgins et al., 2018) is a promising path to facilitate reliable generalization to in- and out-of-distribution downstream tasks (Bengio et al., 2013; Higgins et al., 2018; Milbich et al., 2020; Dittadi et al., 2021; Horan et al., 2021), on top of being more interpretable and fair (Locatello et al., 2019a; Träuble et al., 2021). While Higgins et al. (2018) propose a formal definition based on group equivariance, and various metrics have been proposed to measure disentanglement (Higgins et al., 2017; Chen et al., 2018; Eastwood & Williams, 2018) the most commonly understood definition is as follows:

**Definition 1.1 (Disentanglement)** *Assuming data generated by a set of unknown ground-truth latent factors, a representation is said to be disentangled if there exists a one-to-one correspondence between each factor and dimension of the representation.*

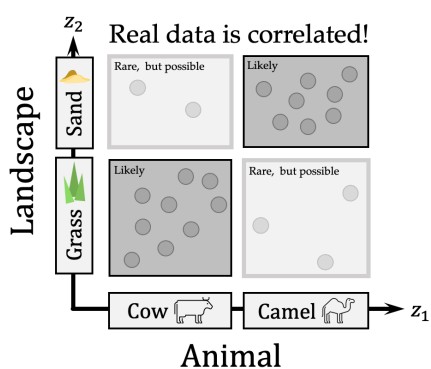

Figure 1: *Real data exhibits correlations* between generative factors: cows are likely on grass, camels on sand. This contradicts disentanglement methods assuming statistically independent factors. Instead, we show that merely assuming and aiming for a factorized support can yield robust disentanglement even under correlated factors.

---

[*]Work done during an internship at Meta AI, FAIR.
[†]Equal contribution

The *method* by which to achieve this goal however, remains an open research question. Weak and semi-supervised settings, e.g. using data pairs or auxiliary variables, can provably offer disentanglement (Bouchacourt et al., 2018; Locatello et al., 2020b; Khemakhem et al., 2020; Klindt et al., 2021). But fully unsupervised disentanglement – our focus in this study – is in theory impossible to achieve in the general unconstrained nonlinear case (Hyvärinen & Pajunen, 1999; Locatello et al., 2019b). In practice however the inductive biases embodied in common autoencoder architectures allow for effective practical disentanglement (Rolinek et al., 2019). Perhaps more problematic, standard unsupervised disentanglement methods (s.a. Higgins et al. (2017); Kim & Mnih (2018); Chen et al. (2018)) rely on an unrealistic assumption of statistical independence of ground truth factors. Real data however contains correlations (Träuble et al., 2021). Even with well defined factors (s.a. shape, color or background), correlations are pervasive—yellow bananas are more frequent than red ones; cows more often on grass than sand. In more realistic settings with correlations, prior work (e.g. Träuble et al. (2021); Dittadi et al. (2021)) has shown existing disentanglement methods to fail.

To address this limitation, we propose to *relax* the unrealistic assumption of statistical independence of factors (i.e. that they have a factorial distribution), and only assume the (bounded) *support* of the factors' distribution factorizes – a much weaker but more realistic constraint. For example, in a dataset of animal images (Fig. 1), background and animal are heavily correlated (camels most likely on sand, cows on grass), resulting in most datapoints being distributed along the diagonal as opposed to uniformly. Under the original assumption of factor independence, a model likely learns a shortcut solution where animal and landscape share the same latent correspondence (Beery et al., 2018). On the other hand with a factorized support, learned factors should be such that any combination of their values has some grounding in reality: a cow on sand is an unlikely, yet not impossible combination. We still rely, just as standard unsupervised disentanglement methods, on the inductive bias of encoder-decoder architectures to recover factors (Rolinek et al., 2019). However, we expect our method to facilitate *robustness to any distribution shifts within the support* (Träuble et al., 2021; Dittadi et al., 2021), as it makes no assumptions on the distribution beyond its factorized support. We arrived at this factorized support principle from the perspective of relaxing the independence assumption to be robust to factor correlations, while remaining agnostic to how they may arise. Remarkably, the same principle was derived independently in Wang & Jordan (2021) [1] from a causal perspective and formal definition of *causal disentanglement* (Suter et al., 2019), that explicits how factor correlations can arise. To ensure a computationally tractable and efficient criterion even with many factors, we *further relax* the full factorized support assumption to that of only a pairwise factorized support, i.e. factorized support for all pairs of factors. On this basis, we propose a concrete pairwise Hausdorff Factorized Support (HFS) training criterion to disentangle correlated factors, by aiming for all pairs of latents to have a factorized support. Specifically we encourage a factorized support by minimizing a Hausdorff set-distance between the finite sample approximation of the actual support and its factorization (Huttenlocher et al., 1993; Rockafellar & Wets, 1998).

Across large-scale experiments on standard disentanglement benchmarks and novel extensions with correlated factors, HFS consistently facilitates disentanglement. We also show that HFS can be implemented as regularizer for other methods to reliably improve disentanglement, up to $+61\%$ in disentanglement performance over baselines as measured by DCI-D (Eastwood & Williams, 2018) (§4.1, Tab. 1). On downstream classification tasks, we improve generalization to more severe distribution shifts and sample efficiency (§4.2, Fig. 2). To summarize our contributions:

**[1]** We motivate and investigate a principle for learning disentangled representations under correlated factors: we relax the assumption of statistically independent factors into that of a factorized support only (independently also derived in Wang & Jordan (2021) from a causal perspective), and further relax it to a more practical *pairwise* factorized support.
**[2]** We develop a concrete training criterion through a pairwise Hausdorff distance term, which can also be combined with existing disentanglement methods (§2.3).
**[3]** Extensive experiments on three main benchmarks and up to 14 increasingly difficult correlations settings over more than 20k models, show HFS systematically improving disentanglement (as measured by DCI-D) by up to $+61\%$ over standard methods ($\beta$/TC/Factor/Annealed-VAE, c.f. §4.1).
**[4]** We show that HFS improves robustness to factor distribution shifts between train and test over disentanglement baselines on classification tasks by up to $+28\%$, as well as sample efficiency.

---

[1] We were initially not aware of this work, whose preprint predates ours. We consider it a strong positive sign when the same principle (of support factorization) is arrived at independently from two quite different angles (causality versus relaxed factor independence assumption).

## 2 PROPOSED APPROACH

### 2.1 DISENTANGLEMENT VERSUS INDEPENDENCE

We are given a dataset $\mathcal{D} = \{\mathbf{x}^i\}_{i=1}^N$ (e.g. images), where each $\mathbf{x}^i$ is a realization of a random variable, e.g., an image. We consider that each $\mathbf{x}^i$ is generated by an unknown generative process, involving a ground truth latent random vector $\mathbf{z}$ whose components correspond to the dataset's underlying factors of variations (s.a. object shape, color, background, . . . ). This process generates an observation $\mathbf{x}$, by first drawing a realization $\mathbf{z} = (z_1, \ldots, z_k)$ from a distribution $p(\mathbf{z})$, i.e. $\mathbf{z} \sim p(\mathbf{z})$. Observation $\mathbf{x}$ is then obtained by drawing $\mathbf{x} \sim p(\mathbf{x}|\mathbf{z})$. Given $\mathcal{D}$, the *goal* of disentangled representation learning can be stated as learning a mapping $f_\phi$ that for any $\mathbf{x}$ recovers as best as possible the associated $\mathbf{z}$ i.e. $f_\phi(\mathbf{x}) \approx \mathbb{E}[\mathbf{z}|\mathbf{x}]$ up to a permutation of elements and elementwise bijective transformation. In unsupervised disentanglement, the $\mathbf{z}$ are unobserved, and both $p(\mathbf{z})$ and $p(\mathbf{x}|\mathbf{z})$ are *a priori unknown to us*, though we might assume specific properties and functional forms. Most unsupervised disentanglement methods follow the formalization of VAEs and employ parameterized *probabilistic generative models* of the form $p_\theta(\mathbf{x}, \mathbf{z}) = p_\theta(\mathbf{z})p_\theta(\mathbf{x}|\mathbf{z})$ to estimate the ground truth generative model over $\mathbf{z}, \mathbf{x}$. As in VAEs, these methods make the strong assumption that ground truth factors are statistically independent:

$$p(\mathbf{z}) = p(z_1)p(z_2) \ldots p(z_k). \tag{1}$$

and conflate the goal of learning a disentangled representation with that of learning a representation with statically independent components. This assumption naturally translates to a factorial model prior $p_\theta(\mathbf{z})$. Successful variants of VAE for disentanglement (Higgins et al., 2017; Kim & Mnih, 2018; Chen et al., 2018) further modify the original VAE objective to even more strongly enforce elementwise independence of the aggregate posterior (i.e. the *encoder* outputs) than afforded by the VAE's optimized evidence lower bound. However, as explained in the introduction, the assumption of factor independence clearly doesn't hold for realistic data distributions. Consequently, methods that enforce this unrealistic assumption suffer from that discrepancy, as shown in Träuble et al. (2021); Dittadi et al. (2021) and confirmed in our own experiments. To address this shortcoming, we develop a novel method to *relax* the unrealistic assumption of factor independence.

### 2.2 RELAXING THE INDEPENDENCE ASSUMPTION INTO THAT OF FACTORIZED SUPPORT

Instead of assuming independent factors (i.e. a factorial distribution on $\mathbf{z}$ as in Eq. 1) we will only assume that the *support* of the distribution factorizes. Let us denote by $\mathcal{S}(p(\mathbf{z}))$ the *support* of $p(\mathbf{z})$, i.e. the set $\{\mathbf{z} \in \mathcal{Z} \,|\, p(\mathbf{z}) > 0\}$. We say that $\mathcal{S}(p(\mathbf{z}))$ is factorized if it equals to the Cartesian product of supports over individual dimensions' marginals, i.e. if:

$$\mathcal{S}(p(\mathbf{z})) = \mathcal{S}(p(z_1)) \times \mathcal{S}(p(z_2)) \times \ldots \times \mathcal{S}(p(z_k)) \stackrel{\text{def}}{=} \mathcal{S}^{\times}(p(\mathbf{z})) \tag{2}$$

where $\times$ denotes the Cartesian product. Of course, 1 (independence) $\Rightarrow$ 2 (factorized-support) but 2 (factorized-support) $\not\Rightarrow$ 1 (independence). Assuming a factorized support is thus a *relaxation* of the (unrealistic) assumption of factorial distribution, (i.e. of statistical independence) of disentangled factors. Refer to the cartoon example in Fig. 1, where the distribution of the two disentangled factors would not satisfy an independence assumption, but does have a factorized support. Informally the factorized support assumption is merely stating that whatever values $z_1$ and $z_2$, etc... may take individually, any combination of these is *possible* (even when not very likely). In the next section we will develop a concrete training criterion that encourages the obtained latent representation to have a factorized support rather than a factorial distribution.

### 2.3 A PRACTICAL CRITERION FOR FACTORIZED SUPPORT

Based on our relaxed hypothesis, we now define a concrete *training criterion* that encourages a factorized support. Let us consider deterministic *representations* obtained by the encoder $\mathbf{z} = f_\phi(\mathbf{x})$. We enforce the factorial support criterion on the aggregate distribution $\bar{q}_\phi(\mathbf{z}) = \mathbb{E}_{\mathbf{x}}[f_\phi(\mathbf{x})]$, where $\bar{q}_\phi(\mathbf{z})$ is conceptually similar to the *aggregate posterior* $q_\phi(\mathbf{z})$ in e.g. TCVAE, though we consider points produced by a deterministic mapping $f_\phi$ rather than a stochastic one. To match our factorized support assumption on the ground truth we want to encourage the support of $\bar{q}_\phi(\mathbf{z})$ to factorize, i.e. that $\mathcal{S}(\bar{q}_\phi(\mathbf{z}))$ and the Cartesian product of each dimension support, $\mathcal{S}^{\times}(\bar{q}_\phi(\mathbf{z}))$, are equal. For

clarity we use shorthand notations $\mathcal{S}$ and $\mathcal{S}^\times$ to denote $\mathcal{S}^\times(\bar{q}_\phi(\mathbf{z}))$ and $\mathcal{S}(\bar{q}_\phi(\mathbf{z}))$ respectively when it is clear from context. To guide the learning, we thus need a divergence or metric to tell us how far $\mathcal{S}$ is from $\mathcal{S}^\times$. Supports are sets, so it is natural to use a set distance such as the Hausdorff distance.

**Hausdorff distance between sets**    Given a base distance metric $d(\mathbf{z}, \mathbf{z}')$ between any two points in $\mathcal{Z}$ (e.g. the Euclidean metric in $\mathcal{Z} = \mathbb{R}^k$), the Hausdorff Distance between sets (here, $\mathcal{S}^\times$ and $\mathcal{S}$), is then defined as

$$d_H(\mathcal{S}, \mathcal{S}^\times) = \max\left(\sup_{\mathbf{z} \in \mathcal{S}^\times}\left[\inf_{\mathbf{z}' \in \mathcal{S}} d(\mathbf{z}, \mathbf{z}')\right], \sup_{\mathbf{z} \in \mathcal{S}}\left[\inf_{\mathbf{z}' \in \mathcal{S}^\times} d(\mathbf{z}, \mathbf{z}')\right]\right) = \sup_{\mathbf{z} \in \mathcal{S}^\times}\left[\inf_{\mathbf{z}' \in \mathcal{S}} d(\mathbf{z}, \mathbf{z}')\right] \quad (3)$$

with the second part of the Hausdorff distance equating to zero since $\mathcal{S} \subset \mathcal{S}^\times$.

**Monte-Carlo Hausdorff Distance Estimation**    In practice we only have a finite sample of observations $\{\mathbf{x}\}_i^N$, and can only estimate the support and Hausdorff distances from the finite number of representations $\{f_\phi(\mathbf{x})\}_i^N$. We thus introduce a practical Monte-Carlo batch-approximation : with access to a batch of $b$ inputs $\mathbf{X}$ yielding $b$ $k$-dimensional latent representations $\mathbf{Z} = f_\phi(\mathbf{X}) \in \mathbb{R}^{b \times k}$, we estimate Hausdorff distances using sample-based approximations to the support: $\mathcal{S} \approx \mathbf{Z}$ and $\mathcal{S}^\times \approx \mathbf{Z}_{:,1} \times \mathbf{Z}_{:,2} \times ... \times \mathbf{Z}_{:,k} = \{(z_1, \ldots, z_k), \; z_1 \in \mathbf{Z}_{:,1}, \ldots, z_k \in \mathbf{Z}_{:,k}\}$. Here $\mathbf{Z}_{:,j}$ must be understood as the *set* (not vector) of all elements in the $j^{\text{th}}$ column of $\mathbf{Z}$. Plugging into Eq. 3 yields:

$$\hat{d}_H(\mathbf{Z}) = \max_{\mathbf{z} \in \mathbf{Z}_{:,1} \times \mathbf{Z}_{:,2} \times ... \times \mathbf{Z}_{:,k}} \left[\min_{\mathbf{z}' \in \mathbf{Z}} d(\mathbf{z}, \mathbf{z}')\right] \quad (4)$$

where by noting $\mathbf{z}' \in \mathbf{Z}$ we consider the matrix $\mathbf{Z}$ as a *set* of rows, over which we find the min[2].

**Further relaxing the assumption to pairwise factorization**    In high dimension, with many factors, the assumption that *every combination of all latent values is possible* might still be too strong an assumption. And even if we assumed all to be in principle possible, we can never hope to observe all in a finite dataset of realistic size due to the combinatorial explosion of conceivable combinations. However, it is statistically reasonable to expect evidence of a factorized support for all pairs of elements[3]. To encourage such a pairwise factorized support, we can minimize a sliced/pairwise Hausdorff estimate with the additional benefit of keeping computation tractable when $k$ is large

$$\hat{d}_H^{(2)}(\mathbf{Z}) = \sum_{i=1}^{k-1} \sum_{j=i+1}^{k} \max_{\mathbf{z} \in \mathbf{Z}_{:,i} \times \mathbf{Z}_{:,j}} \left[\min_{\mathbf{z}' \in \mathbf{Z}_{:,(i,j)}} d(\mathbf{z}, \mathbf{z}')\right] \quad (5)$$

where $\mathbf{Z}_{:,(i,j)}$ denotes the concatenation of column $i$ and column $j$, yielding a *set of rows*.

**Avoiding collapse and retaining input information**    We will be learning representations $\mathbf{z} = f_\phi(\mathbf{x})$ by learning parameters $\phi$ that optimize a training objective. Because the Hausdorff distance builds on a base distance $d(\mathbf{z}, \mathbf{z}')$, if we were to minimize only this, it could be trivially minimized to 0 by collapsing all representations to a single point. Avoiding this can be achieved in several ways, s.a. by including a term that encourages the variance of $\mathbf{z}_{:,i}$ to be above 1 (a technique used e.g. in self-supervised learning method VICReg (Bardes et al., 2022)) or – more in line with traditional VAE variants for disentanglement – by using a stochastic autoencoder (SAE) reconstruction error:

$$\ell_{\text{SAE}}(\mathbf{x}; \phi, \theta) = -\mathbb{E}_{q_\phi(\mathbf{z}|\mathbf{x})} \left[\log p_\theta(\mathbf{x}|\mathbf{z})\right] \quad (6)$$

where typically $q_\phi(\mathbf{z}|\mathbf{x}) = \mathcal{N}(f_\phi(\mathbf{x}), \Sigma_\phi(\mathbf{x}))$ with mean given by our deterministic mapping $f_\phi$, $\Sigma_\phi(\mathbf{x})$ producing a diagonal covariance parameter, and e.g. $\log p_\theta(\mathbf{x}|\mathbf{z}) = \|r_\theta(z) - x\|^2$ with $r_\theta$ a parameterized decoder. The autoencoder term ensures representations $f_\phi(\mathbf{x})$ retaining as much information as possible about $\mathbf{x}$ for reconstruction, preventing collapse of representations to a single point. A minimum scale can also be ensured by imposing $\Sigma_\phi(\mathbf{x})$ to be above a minimal threshold.

## 2.4 PUTTING IT ALL TOGETHER

Our basic training objective for Hausdorff-based Factored Support (HFS) can thus be formed by simply combining the stochastic auto-encoder loss of Eq. 6 and our Hausdorff estimate of Eq. 5:

$$\mathcal{L}_{\text{HFS}}(\mathcal{D}; \phi, \theta) = \mathbb{E}_{\mathbf{X} \overset{b}{\sim} \mathcal{D}} \left[\gamma \hat{d}_H^{(2)}(f_\phi(\mathbf{X})) + \tfrac{1}{b} \sum_{\mathbf{x} \in \mathbf{X}} \ell_{\text{SAE}}(\mathbf{x}; \phi, \theta)\right] \quad (7)$$

---

[2]One can alternatively use softened max and min operations, as defined in Appendix A.4. In practice, we saw no robustness benefit to this, likely because we compute $\hat{d}_H$ over batches, not the entire dataset.

[3]Straightforward to generalize to larger tuples, but computational and statistical benefits shrink accordingly.

where $\mathbf{X} \overset{b}{\sim} \mathcal{D}$ denotes a batch of $b$ inputs, $f_\phi(\mathbf{X})$ the batch representations $\mathbf{Z}$, and $\gamma$ the trade-off between the Hausdorff and SAE terms. To compare with existing VAE-based disentanglement methods s.a. $\beta$-VAE (Higgins et al., 2017), we can also use Eq. 5 as regularizer on top:

$$\mathcal{L}_{\mathrm{HFS}}^{\beta\mathrm{VAE}}(\mathcal{D}; \phi, \theta) = \mathbb{E}_{\mathbf{X} \overset{b}{\sim} \mathcal{D}}\left[\gamma \hat{d}_H^{(2)}(\mathbf{Z}) + \frac{1}{b} \sum_{\mathbf{x} \in \mathbf{X}} \left(\ell_{\mathrm{SAE}}(\mathbf{x}; \phi, \theta) + \beta D_{\mathrm{KL}}(q_\phi(\mathbf{z}|\mathbf{x}) || p_\theta(\mathbf{z}))\right)\right] \quad (8)$$

where $D_{\mathrm{KL}}$ is the Kullback-Leibler divergence, and $p_\theta(\mathbf{z})$ the usual VAE factorial unit Gaussian prior. This hybrid objective recovers the original $\beta$-VAE with $\gamma = 0$, and $\mathcal{L}_{\mathrm{HFS}}$ (Eq. 7) with $\beta = 0$, showing that the plain HFS objective *replaces* the $\beta$-VAE KL term by our factorized-support-encouraging Hausdorff term and removes the factorial prior $p(\mathbf{z})$. We can similarly extended other VAE-based variants (Chen et al., 2018; Kim & Mnih, 2018; Burgess et al., 2018) by adding our Hausdorff term as regularizer to focus more on its support than a precise factorial distribution.

## 3 RELATED WORK

**Disentangled Representation Learning** aims to recover representation spaces where each ground-truth generative factor is encoded in a unique entry or subspace (Bengio et al., 2013; Higgins et al., 2018) to benefit subsequent downstream transfer (Bengio et al., 2013; Peters et al., 2017; Tschannen et al., 2018; Locatello et al., 2019b; Montero et al., 2021; Mancini et al., 2021; Roth et al., 2020; Funke et al., 2022), interpretability (Chen et al., 2016; Esser et al., 2018; Niemeyer & Geiger, 2021) and fairness (Locatello et al., 2019a; Träuble et al., 2021; Dullerud et al., 2022) via compositionality of representations. Methods often rely on Variational AutoEncoders (VAEs) variants (Kingma & Welling, 2014; Rezende et al., 2014) to constrain the (aggregate) posterior of the encoder, e.g. via penalties on the bottleneck capacity ($\beta$-VAE (Higgins et al., 2017)) with progressive constraints or network growing (AnnealedVAE (Burgess et al., 2018), ProVAE (Li et al., 2020)), the total correlation ($\beta$-TCVAE (Chen et al., 2018), FactorVAE (Kim & Mnih, 2018)) or the mismatch to some factorized prior (DIP-VAE (Kumar et al., 2018), DoubleVAE (Mita et al., 2020)). These approaches assume *statistically independent* factors, which is invalid for realistic data as motivated in §1.

**Disentanglement under correlated factors.** Consequently, while most methods have been shown to perform well on toy datasets and ones with known independent factors such as Shapes3D (Kim & Mnih, 2018), MPI3D (Gondal et al., 2019), DSprites (Higgins et al., 2017), SmallNorb (LeCun et al., 2004) or Cars3D (Reed et al., 2015), recent research (Montero et al., 2021; Träuble et al., 2021; Montero et al., 2022; Funke et al., 2022; Dittadi et al., 2021) has started to connect these setups to more realistic scenarios with factor correlations: Träuble et al. (2021) introduce artificial correlations between two factors, and Montero et al. (2021) exclude value combinations for recombination studies. In such settings, Montero et al. (2021; 2022); Träuble et al. (2021); Dittadi et al. (2021) show that unsupervised disentanglement methods that assume independent factors fail to disentangle, with potentially negative impact to OOD generalization. Suter et al. (2019) propose a causal metric to evaluate disentanglement when assuming confounders between the ground-truth factors. Choi et al. (2020) introduce a Gaussian mixture model for dependencies between continuous and discrete variables in a structured setup with number of mixtures known. By contrast, we investigate a generic remedy without explicit auxiliary variables or prior models by relaxing the independence assumption to only a *pairwise factorized support*. Pfau et al. (2020) propose geometrically motivated non-parametric unsupervised disentanglement following the symmetry-based definition in Higgins et al. (2018) by leveraging holonomy of manifold geometries as learning signal to find disentangled subspaces. This does not assume statistical independence, but requires non-trivial holonomy for each factor manifold, and struggles in high-dimensional spaces and generalization to new data. For domain adaptation shifts, Tong et al. (2022) propose adversarial support matching, highlighting that operating on the support can be beneficial for related settings as well. To evaluate disentanglement, we utilize DCI-D (part of DCI – **D**isentanglement, **C**ompleteness, **I**nformativeness, see Eastwood & Williams (2018)) as leading metric. As opposed to other metrics s.a. Beta-/FactorVAE scores (Higgins et al., 2017; Kim & Mnih, 2018), MI Gap (Chen et al., 2018), Modularity (Ridgeway & Mozer, 2018) or SAP (Kumar et al., 2018), Locatello et al. (2020a); Dittadi et al. (2021) have indicated DCI-D as the potentially most suitable disentanglement metric (and as also done e.g. in Locatello et al. (2019b; 2020b); Träuble et al. (2021)), with generally strong correlation between metrics (Locatello et al., 2019b). Finally, Wang & Jordan (2021) independently also arrived at the idea of support factorization for disentanglement from a causal perspective, for which they propose a similar Hausdorff distance objective, providing orthogonal validation to support factorization for disentanglement. On

the contrary, we derive it from relaxing the assumption of factor independence, and propose further pairwise relaxation, which performs and scales much better (see Supp. A), alongside a much more expansive experimental study on the impact on downstream disentanglement, adaptation and generalization under various correlation shifts.

## 4 EXPERIMENTS

We start with experimental details listed below, before studying HFS on benchmarks with and without training correlations (§4.1). These results are extended in §4.2 to evaluate the transfer and downstream adaptability (§4.3) of learned representations across different correlation shifts and link HFS during training to various downstream metrics (§4.4). We include variant, qualitative and hyperparameter robustness studies in appendix §B, §E and §D - all favouring our HFS objective. Across experiments, we re-implemented baselines ($\beta$-VAE (Higgins et al., 2017), FactorVAE (Kim & Mnih, 2018), AnnealedVAE (Burgess et al., 2018), $\beta$-TCVAE (Chen et al., 2018)) as done e.g., in Locatello et al. (2019b; 2020b); Träuble et al. (2021). To investigate methods under correlated ground truth factors, we use and extend the correlation framework introduced in Träuble et al. (2021) who introduce correlation between pairs of factors as $p(z_1, z_2) \propto \exp\left(-(z_1 - f(z_2))^2/(2\sigma^2)\right)$, where higher $\sigma$ notes weaker correlation between normalized factors $z_1$ and $z_2$, and $f(z) = z$ or $f(z) = 1 - z$ for inverted correlations when necessary . We extend this framework to include correlations between *multiple* factor pairs (either 1, 2 or 3 pairs) and shared confounders (one factor correlated to all others). All reported numbers are computed on at least 6 seeds (with $\geq 10$ seeds used for key experiments s.a. Tab. 1 or Fig. 2). Similar to existing literature (Locatello et al., 2019b; 2020b; Träuble et al., 2021; Dittadi et al., 2021) we cover at least 7 hyperparameter settings for each baseline. Further experimental details are provided in §H.

### 4.1 FACTORIZATION OF SUPPORTS FOR DISENTANGLEMENT ON STANDARD BENCHMARK

We study the behaviour of HFS and baselines on standard disentanglement learning benchmarks and correlated variants thereof (see §4) - Shapes3D (Kim & Mnih, 2018), MPI3D Gondal et al. (2019) and DSprites (Higgins et al., 2017). For each setting, we report results averaged over $\geq 10$ seeds in Tab. 1. Each column denotes the test performance on uncorrelated data for all methods trained on a particular correlation setting. As DSprites only has five effective factors of variation, no three-pair setting is possible. Values reported denote median DCI-D with 25th and 75th percentiles in grey. Our results indicate that a factorization of the support via HFS encourages disentanglement (as measured via DCI-D) without relying on a factorial distribution objective (s.a. $\beta$-VAE and its variants), consistently matching or outperforming the comparable $\beta$-VAE setting - both when no correlation is encountered during training (*"No Corr."*) as well as for much more severe correlations (*"Conf."*). Even more, we find that extending existing disentanglement objectives s.a. $\beta$-VAE (or stronger extensions like $\beta$-TCVAE) with explicit factorization of the support (+HFS) can provide even further, **significant** improvements. For example, without correlations we find relative improvements of nearly **+30%**, while for some correlated settings, e.g. with a shared confounder, these go over **+60%**! In addition, relative increases of up to **+140%** on $\beta$-TCVAE further highlight both the general importance of an explicit factorization of the support for disentanglement even under training correlations, as well as it being a property generally neglected until now.

### 4.2 OUT-OF-DISTRIBUTION GENERALIZATION UNDER CORRELATION SHIFTS

As $\beta$-VAE and $\beta$-VAE + HFS models in §4.1 were trained on correlated and evaluated on uncorrelated data, the performance differences provide a first indication that encouraging a factorized support can benefit transfer under correlation shifts. Such changes in correlation from train to test data is commonly referred to as "distributional shift" (Quinonero-Candela et al., 2009) as the test data becomes out-of-distribution for the model, and mark a key issue interfering with generalization in realistic settings (Arjovsky et al., 2019; Koh et al., 2021; Milbich et al., 2021; Roth et al., 2022; Funke et al., 2022). While some works point to initial benefits of disentangled representations for out-of-distribution (OOD) generalization (e.g. Träuble et al. (2021); Dittadi et al. (2021)), some have raised concerns about the gains from disentanglement e.g. on OOD recombination (Montero et al., 2022). Conceptually, one way a disentangled representation can benefit a downstream prediction task is when the true predictor is a function of only a *subset* of the true factors. Successful recovery

Table 1: *Disentanglement by explicitly factorizing the support* using HFS on 3 benchmarks across various numbers of correlated factors (columns) and correlation increasing from left (no correlation) to right (every factor correlated to one confounder; for DSprites, three pairs are impossible, see text). Scores denote DCI-D metric computed on uncorrelated test data. (**Bold**) blue denotes (second) best performance per benchmark/correlation. [a, b] indicate 25/75th percentiles. The results show that relaxing the goal of a factorial latent distribution to a **factorized support** with standalone HFS already **offers competitive disentanglement**. Adding HFS as regularizer over standard methods ($\beta$-VAE/TCVAE) to target a more factorized support yields even higher scores, beating other approaches with optimally tuned hyperparameters s.a. $\beta$. Remarkably on MPI3D, optimal tuning turned $\beta$ and other TCVAE terms to 0, leaving only HFS which consistently worked best.

| METHOD | NO CORR. | PAIRS: 1 $\sigma = 0.1$ | PAIRS: 2 $\sigma = 0.1$ | PAIRS: 3 $\sigma = 0.1$ | SHARED CONF. $\sigma = 0.2$ |
|---|---|---|---|---|---|
| **Shapes3D** (Kim & Mnih, 2018) | | | | | |
| $\beta$-VAE | 70.7 [65.2, 75.4] | 71.6 [60.9, 72.5] | 55.6 [45.8, 57.2] | 37.1 [32.3, 38.9] | 38.0 [35.0, 38.6] |
| HFS | 78.3 [75.1, 83.6] | 77.8 [74.4, 78.8] | 56.0 [41.0, 57.4] | 47.5 [38.0, 49.1] | 46.2 [36.3, 47.5] |
| $\beta$-VAE + HFS | **91.2 [75.8, 100.0]** | **80.9 [76.4, 81.4]** | **67.6 [62.4, 69.3]** | **47.9 [44.0, 50.8]** | **63.5 [61.2, 65.5]** |
| $\beta$-TCVAE | 77.1 [76.6, 78.3] | 71.1 [65.5, 72.5] | 63.8 [59.1, 65.1] | 47.3 [36.7, 50.0] | 49.9 [45.8, 55.9] |
| $\beta$-TCVAE + HFS | **85.7 [82.5, 97.3]** | **75.2 [63.1, 76.6]** | **68.3 [61.0, 71.8]** | **51.6 [47.7, 52.8]** | **61.5 [53.8, 64.2]** |
| FactorVAE | 66.1 [51.2, 69.1] | 70.8 [70.5, 71.2] | 57.2 [55.9, 62.0] | 46.8 [40.8, 49.0] | 31.6 [27.9, 35.1] |
| AnnealedVAE | 62.2 [60.7, 63.2] | 57.2 [49.5, 59.3] | 31.6 [26.9, 34.1] | 33.6 [31.0, 38.0] | 23.0 [20.1, 25.9] |
| **DSprites** (Higgins et al., 2017) | | | | | |
| $\beta$-VAE | 32.2 [25.3, 37.9] | 9.5 [7.9, 10.3] | 7.5 [6.7, 8.3] | N/A | 11.4 [9.9, 13.9] |
| HFS | 34.9 [27.4, 36.0] | 13.6 [7.6, 16.7] | 11.9 [9.7, 13.8] | N/A | 15.1 [11.0, 16.0] |
| $\beta$-VAE + HFS | **49.9 [30.0, 50.4]** | 19.7 [17.0, 21.1] | **17.3 [6.0, 19.6]** | N/A | 15.8 [12.3, 16.7] |
| $\beta$-TCVAE | 30.9 [28.9, 35.2] | 24.0 [23.6, 24.4] | 11.4 [7.6, 13.6] | N/A | **20.9 [17.5, 23.6]** |
| $\beta$-TCVAE + HFS | **53.1 [41.8, 53.2]** | **26.5 [25.6, 27.2]** | **27.8 [16.1, 31.6]** | N/A | **24.8 [23.8, 26.3]** |
| FactorVAE | 25.7 [20.9, 30.9] | 15.1 [11.9, 16.3] | 13.4 [12.4, 15.0] | N/A | 14.7 [13.5, 15.3] |
| AnnealedVAE | 39.4 [38.7, 40.0] | 14.8 [14.3, 15.9] | 8.5 [6.9, 10.3] | N/A | 14.3 [14.1, 14.5] |
| **MPI3D** (Gondal et al., 2019) | | | | | |
| $\beta$-VAE | 25.6 [24.7, 26.1] | 20.5 [17.7, 20.9] | 23.6 [22.6, 24.3] | 11.6 [11.1, 11.7] | 11.8 [10.0, 12.7] |
| HFS | **32.8 [30.0, 34.3]** | **28.4 [26.5, 29.5]** | **28.0 [27.4, 28.2]** | **14.3 [13.1, 14.8]** | **16.1 [15.0, 16.6]** |
| $\beta$-VAE + HFS | **32.8 [30.0, 34.3]** | **28.4 [26.5, 29.5]** | **28.0 [27.4, 28.2]** | **14.3 [13.1, 14.8]** | **16.1 [15.0, 16.6]** |
| $\beta$-TCVAE | 26.6 [26.0, 27.4] | 20.7 [20.4, 21.3] | 23.3 [21.9, 23.8] | 11.4 [10.3, 12.6] | 14.2 [13.4, 15.4] |
| $\beta$-TCVAE + HFS | **32.8 [30.0, 34.3]** | **28.4 [26.5, 29.5]** | **28.0 [27.4, 28.2]** | **14.3 [13.1, 14.8]** | **16.1 [15.0, 16.6]** |
| FactorVAE | 26.0 [24.8, 27.5] | 21.9 [20.1, 23.9] | 27.8 [27.2, 29.2] | 10.9 [10.7, 11.9] | 13.6 [12.8, 13.9] |
| AnnealedVAE | 11.8 [10.8, 12.4] | 11.7 [10.4, 12.9] | 11.8 [11.6, 12.1] | 11.6 [10.6, 12.2] | 13.4 [12.8, 13.9] |

of all factors, disentangled, enables effective subsequent feature selection (e.g. L1-regularized logistic regression or shallow decision trees). **A downstream predictor can thus be far more sample efficient (Ng, 2004) in learning to ignore irrelevant factors**, that may be spuriously correlated with the target, than if they were entangled in the representation. As we showed that explicit support factorization provides stronger relative disentanglement, we leverage this for further insights into its benefits on OOD tasks. Given the strong performance of HFS on Shapes3D, we extend our experiments from §4.1 on this dataset with more training and now also test data correlations. This gives transfer grids (Fig. 2) across diverse, increasingly severe correlation shifts. For each grid, the y- and x-axis indicate training and test correlations increasing from top to bottom and left to right, respectively. Darker colors refer to a score increase. We use these grids to see **(1)** how different correlation shifts impact disentanglement, and **(2)** if improvements in disentanglement via explicitly aiming for a factorized support impact downstream transferability of the learned representations.

**(1)** We first evaluate disentanglement of a standard $\beta$-VAE (**leftmost** grid; each square uses optimal parameters for a given correlation and seed), and find an expected drop with increased correlation on the training data. The subsequent grid shows changes when adding HFS to $\beta$-VAE, with consistent improvements in disentanglement of test data over $\beta$-VAE across **all** correlation shifts (only positive changes), and extends our insights from Tab. 1.

**(2)** To understand the usefulness for practical transfer tasks under correlation shifts, we train a Gradient Boosted Tree (GBT, `sklearn` (Pedregosa et al., 2011), c.f. Locatello et al. (2019b; 2020b)) to

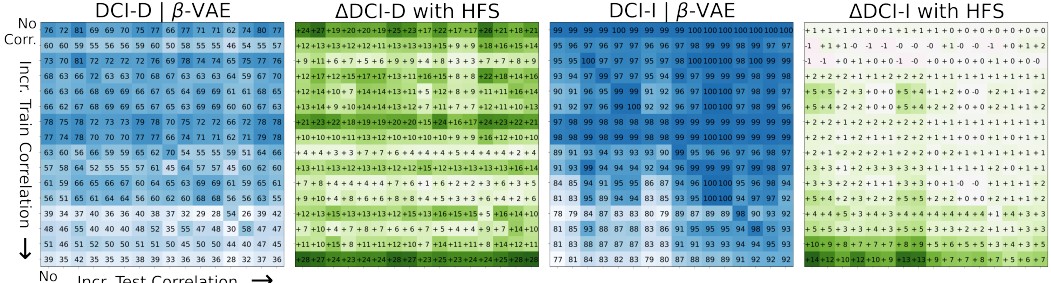

Figure 2: *Out-of-Distribution Disentanglement and Generalization across large ranges of correlation shifts* between train and test data on Shapes3D. We evaluate the impact of encouraging factorized support on disentanglement (DCI-D) and classification performance of test ground truth factors (DCI-I) via HFS. Y-axis denotes source correlations increasing from top to bottom, x-axis target correlations (left to right). Darker blue and green mean higher scores and absolute improvements, respectively. **[Leftmost]:** DCI-D $\beta$-VAE for all shifts, dropping with higher training correlations. **[Left]:** Consistent and in parts high improvements in DCI-D when explicitly encouraging factorized support via $\beta$-VAE + HFS across shifts. **[Right]:** DCI-I using a GBT over embeddings generated by a $\beta$-VAE model trained on respective source correlations. Drop in performance with higher training correlation or test data variation (bottom left corner). **[Rightmost]:** Absolute changes in DCI-I with HFS reveal higher generalization particularly when shifts are large (c.f. bottom-left). This shows that explicitly encouraging factorized support benefits generalization as shifts become more severe.

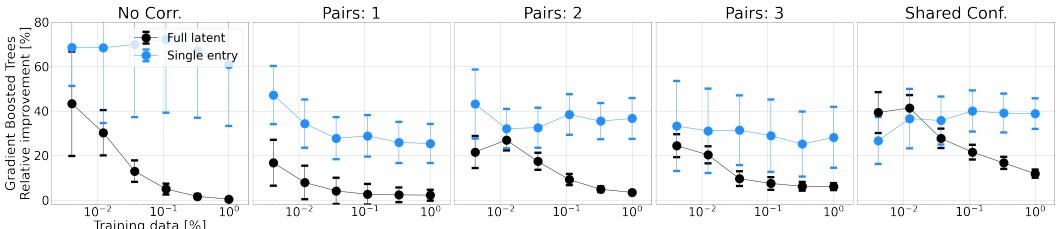

Figure 3: *Increased accuracy and sample efficiency on downstream classification.* We plot relative improvement (%) in average ground truth factor classification accuracy by using HFS on top of a $\beta$-VAE, as a function of the amount of labeled training data. Classifier is a GBT (for linear probe see §F.2) receiving either the entire latent vector (black) or only the most expressive entry (blue). The increased disentanglement through HFS gives consistent improvements in all cases, and gets more pronounced in the low data regime for full latents, **indicating higher sample efficiency, as expected from better disentanglement**. Relative improvements up to $+80\%$ in the single entry case across correlation shifts highlight the better reflection of ground truth factors across correlations.

take representations of the test data and predict the exact ground truth factor values; reflected in the third grid for $\beta$-VAE baseline and measured by the DCI-I metric (Eastwood & Williams, 2018). We see that the downstream classification performance, while saturated for small shifts, drops notably both with increased training correlation or more variation in the test data (drop towards bottom-left corner). If we now measure the change in downstream classification performance **with** HFS (last grid), we see that while for small shifts the benefits are small, they increase for larger ones (increase towards same bottom left corner). This indicates that changes in disentanglement through our support factorization become increasingly important as distributional shifts increase, and highlight that benefits for OOD generalization drawn from improvements in disentanglement may be particularly evident for harder shifts.

### 4.3 BENEFITS UNDER VARYING DOWNSTREAM ADAPTATION METHODS

This section investigates how generalization improvements hold when the amount of downstream test data changes, and revisits the recovery of ground truth factors under correlation shifts by looking at the performance with only the single most important latent entry. We train GBTs (as we care about relative changes, we use xgboost (Chen & Guestrin, 2016) for faster training) on embeddings ex-

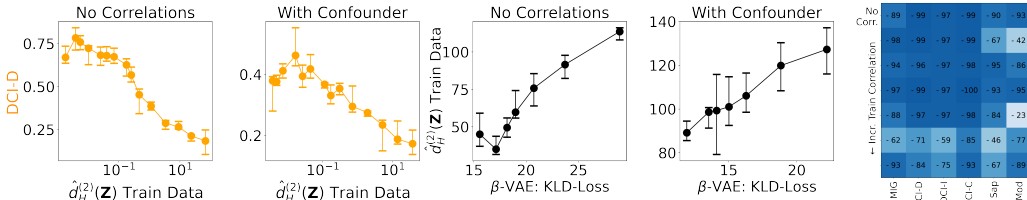

Figure 4: **[Left, orange]**: Increased support factorization on train data measured by lower Hausdorff estimate $\hat{d}_H^{(2)}$ by increasing HFS weight $\gamma$ (Eq. 7)), improves disentanglement (DCI-D) across correlations on Shapes3D (more results in Supp. Fig. 9). **[Center, black]**: Minimizing $\beta$-VAE KL-Divergence by increasing $\beta$ implicitly encourages a factorized support by pushing towards full independence, but hurts disentanglement because of the incorrect assumption. **[Right]**: We find strong correlation between HFS on train data and standard disentanglement metrics on test data ([%], darker $\rightarrow$ higher) even under training correlations (top to bottom). Detailed figure in §F.5.

tracted either from an optimal $\beta$-VAE or $\beta$-VAE + HFS. For the single-latent-entry training, we first train a GBT to select the most important entry to predict each respective ground truth factor, and then use said entry to train a second ground truth factor predictor. See Supp. §F.2 for experiments with linear probes. In all cases (Fig. 3), explicit support factorization via HFS facilitates downstream adaptation with particular benefits when only little data is provided at test time. For example in the standard uncorrelated setting, relative improvements increase from $4\%$ to $45\%$, with similar trends across correlation shifts. Finally, our experiments reveal that increased disentanglement expectedly results in a better reflection of ground truth factors in single latent entries, shown in nearly $+80\%$ relative improvement when training and predicting on the most expressive entry. These insights reinforce that explicit support factorization via HFS encourages disentanglement also under correlation shifts, and show potential benefits in downstream generalization especially in the low data regime.

## 4.4 Factorization as a Performance Metric

We now explore the relationship of HFS to existing metrics across correlations. Utilizing Eq. 5 as separate evaluation metric for the factorization of the support across the whole training data facilitated through increased HFS weighting $\gamma$, we find that when the factorization of the support across the training data goes down (Fig. 4 *(left, orange)*), the disentanglement on the test data consistently goes up, verifying again the connection between support factorization and disentanglement. Fig. 4 *(center, black)* shows $\beta$-VAE implicitly encouraging a factorized support by pushing towards independence, but which hurts disentanglement and generalization, see experiments above. Finally, Fig. 4 *(right)* shows that support factorization on the training data exhibits correlation with disentanglement metrics, consistent across also stronger training correlations, albeit lower. This is useful, as HFS neither requires access to ground truth factors nor a specific prior distribution over the support, and can thus serve as a proxy for development and training evaluation of future works.

## 5 Conclusion

To avoid the unrealistic assumption of factors independence (i.e. factorial distribution) as in traditional disentanglement, which stands in contrast to realistic data being correlated, we thoroughly investigate an approach that only aims at recovering a factorized *support*. Doing so achieves disentanglement by ensuring the model can encode many possible combinations of generative factors in the learned latent space, while allowing for arbitrary distributions over the support – in particular those with correlations. Indeed, through a practical criterion using pairwise Hausdorff set-distances – HFS – we show that encouraging a pairwise factorized support is sufficient to match traditional disentanglement methods. Furthermore we show that HFS can steer existing disentanglement methods towards a more factorized support, giving large relative improvements of over $+60\%$ on common benchmarks across a large variety of correlation shifts. We find this improvement in disentanglement across correlation shifts to be also reflected in improved out-of-distribution generalization especially as these shifts become more severe; tackling a key promise for disentangled representation learning.

## REPRODUCIBILITY STATEMENT

To reproduce the results from this paper and avoid implementational and library-related differences, we have released our codebase here: https://github.com/facebookresearch/disentangling-correlated-factors.

To reproduce Tab. 1, we first refer to Tab. 5, which contains all Tab. 1 results with additional details on the exact correlations used (as well as other correlation settings). For each of the correlation settings, the associated factor correlation pairs are provided in §H.1, with the training, model as well as grid-search details all noted in §H. The correlation formula to introduce artificial correlations between respective factors follows the setup described in the experimental details noted at the beginning of §4.

For the correlation shift transfer experiments used in §4.2, the same training and correlation settings are used. For our downstream adaptability results, we provide all relevant details in §4.3 and §H.

## ACKNOWLEDGEMENTS

The authors would like to thank Léon Bottou for useful and encouraging early discussions, as well as David Lopez-Paz, Mike Rabbat, and Badr Youbi Idrissi for their careful reading and feedback that helped improve the paper. We also want to extend special thanks to Kartik Ahuja for later making us aware of the work of Wang & Jordan (2021) and its close connection with our approach. Karsten Roth thanks the International Max Planck Research School for Intelligent Systems (IMPRSIS) and the European Laboratory for Learning and Intelligent Systems (ELLIS) PhD program for support. Zeynep Akata acknowledges partial funding by the ERC (853489 - DEXIM) and DFG (2064/1 – Project number 390727645) under Germany's Excellence Strategy.

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

# A    ALTERNATE HAUSDORFF VARIANTS

In this section, we introduce various variants to our Hausdorff distance approximation introduced in §2.3 and particularly Eq. 5, which we then experimentally evaluate in §B.

## A.1    AVERAGED HAUSDORFF

First, due to the sensitivity to outliers, one can also utilize the average Hausdorff distance, which simply gives:

$$\hat{d}^{(2)}_{H,avg}(\mathbf{Z}) = \sum_{i=1}^{k} \sum_{j=i+1}^{k} \frac{1}{|\mathbf{Z}_{:,i} \times \mathbf{Z}_{:,j}|} \sum_{\mathbf{z} \in \mathbf{Z}_{:,i} \times \mathbf{Z}_{:,j}} \left[ \min_{\mathbf{z}' \in \mathbf{Z}_{:,(i,j)}} d(\mathbf{z}, \mathbf{z}') \right] \tag{9}$$

using the same pair-based approximation introduced in Eq. 5.

## A.2    SUBSAMPLING

One can also operate on the full approximated $\hat{\mathcal{S}}^{\times}$ with $\mathcal{S}^{\times} \approx \hat{\mathcal{S}}^{\times} = \mathbf{Z}_{:,1} \times \mathbf{Z}_{:,2} \times ... \times \mathbf{Z}_{:,k}$ instead of a collection of $\hat{\mathcal{S}}^{\times}_{i,j} = \mathbf{Z}_{:,i} \times \mathbf{Z}_{:,j}$, by simply utilising a randomly subsampled version of $\hat{\mathcal{S}}^{\times}$, denoted $\hat{\mathcal{S}}^{\times}_{\text{sub}}$ (through i.i.d. stitching of latent entries sampled from each dimension support):

$$\hat{d}_{H,\text{sub}} = \max_{\mathbf{z} \in \hat{\mathcal{S}}^{\times}_{\text{sub}}} \left[ \min_{\mathbf{z}' \in \mathbf{Z}} d(\mathbf{z}, \mathbf{z}') \right] \tag{10}$$

However, in practice (as shown in §B), we found $\hat{d}^{(2)}_H$ to work better, as the max-operation over a collection of 2D subspaces provides a less sparse training signal than a single backpropagated distance pair in $\hat{d}_{H,\text{sub}}$.

## A.3    SAMPLING-BASED SOFTMIN

In addition, as the latent representations and the corresponding support change during training, one can also encourage some degree of exploration during training instead of relying on the use of hard max and min operations, for example through a probabilistic selection of the final distance to minimize for, allowing for a controllable degree of exploration during training:

$$\hat{d}^{(2)}_{H,\text{prob}} = \sum_{i=1}^{k} \sum_{j=i+1}^{k} \max_{\mathbf{z} \in \hat{\mathcal{S}}^{\times}(Z)} \left[ \mathbb{E}_{\mathbf{z}' \sim p_{\text{softmin}}(\cdot | \mathbf{z}, \mathbf{Z}_{:,(i,j)}, \tau)} [d(\mathbf{z}, \mathbf{z}')] \right] \tag{11}$$

with the SoftMin-distribution

$$p_{\text{softmin}}(\mathbf{z}' | \mathbf{z}, \mathbf{Z}, \tau) = \frac{\exp(-d(\mathbf{z}, \mathbf{z}')/\tau)}{\sum_{\mathbf{z}^* \in \mathbf{Z}} \exp(-d(\mathbf{z}, \mathbf{z}^*)/\tau)} \tag{12}$$

though as shown in the following experimental section, we found minimal benefits in doing so.

## A.4    SOFTENED HAUSDORFF DISTANCE

To potentially better align the Hausdorff distance objective with the differentiable optimization process, it may make be beneficial to look into soft variants to relax the hard minimization and maximization, respectively (note the $\tilde{d}$ instead of $\hat{d}$):

$$\sigma_{\min}(\mathbf{z}, \mathbf{z}', \mathbf{Z}) = \frac{\exp(-d(\mathbf{z}, \mathbf{z}')/\tau_1)}{\sum_{\mathbf{z}^* \in \mathbf{Z}} \exp(-d(\mathbf{z}, \mathbf{z}^*)/\tau_1)}$$

$$d^{\text{soft}}_{\min}(\mathbf{z}, \mathbf{Z}) = \sum_{\mathbf{z}' \in \mathbf{Z}} \sigma_{\min}(\mathbf{z}, \mathbf{z}', \mathbf{Z}) \, d(\mathbf{z}, \mathbf{z}') \tag{13}$$

$$\tilde{d}^{(2)}_H(\mathbf{Z}) = \sum_{i=1}^{k} \sum_{j=i+1}^{k} \sum_{\mathbf{z} \in \mathbf{Z}_{:,(i,j)}} \frac{\exp(d^{\text{soft}}_{\min}(\mathbf{z}, \mathbf{Z}_{:,(i,j)})/\tau_2)}{\sum_{\mathbf{z}^* \in \mathbf{Z}_{:,(i,j)}} \exp(d^{\text{soft}}_{\min}(\mathbf{z}^*, \mathbf{Z}_{:,(i,j)})/\tau_2)} d^{\text{soft}}_{\min}(\mathbf{z}, \mathbf{Z}_{:,(i,j)})$$

Table 2: Method ablations to $\hat{d}_H^{(2)}$. We compare our default pairwise Hausdorff approximation against a variant using averaging instead of $\max$ ($\hat{d}_{H,avg}^{(2)}$), a probabilistic approximation to $\min$ ($\hat{d}_{H,prob}^{(2)}$) as well as subsampling of the full-dimensional factorized support without any pairwise approximations. In all cases, entries are selected with at least 5 seeds and optimal values chosen from a gridsearch over $\gamma \in \{1, 3, 10, 30, 100, 300, 1000, 3000, 10000\}$.

| Setup ↓ | No Correlation | Correlated Pairs: 1 | Correlated Pairs: 3 |
|---|---|---|---|
| $\hat{d}_H^{(2)}$ | **76.9** [74.3, 81.4] | **55.9** [47.6, 59.9] | **48.0** [39.1, 48.9] |
| $\hat{d}_{H,avg}^{(2)}$ (Eq. 9) | **61.1** [55.6, 65.5] | **33.5** [26.9, 36.0] | **42.4** [38.9, 46.3] |
| $\hat{d}_{H,prob}^{(2)}$ (Eq. 11) | **74.7** [69.5, 77.4] | **56.2** [46.0, 60.4] | **46.9** [37.7, 48.0] |
| $\hat{d}_H^{sub}$ (Eq. 10), Subs. 80 | **56.4** [52.6, 61.7] | **38.1** [21.2, 42.9] | **29.7** [19.7, 32.0] |
| $\hat{d}_H^{sub}$ (Eq. 10), Subs. 800 | **60.0** [55.0, 62.8] | **44.5** [25.7, 48.0] | **31.6** [24.4, 34.1] |
| $\hat{d}_H^{sub}$ (Eq. 10), Subs. $8 \cdot 10^3$ | **62.2** [59.5, 69.3] | **52.1** [48.8, 54.7] | **32.1** [27.0, 33.1] |
| $\hat{d}_H^{sub}$ (Eq. 10), Subs. $8 \cdot 10^4$ | **66.9** [61.3, 71.2] | **54.5** [49.3, 59.7] | **39.2** [31.9, 43.8] |
| $\hat{d}_H^{sub}$ (Eq. 10), Subs. $8 \cdot 10^5$ | **66.6** [62.2, 72.4] | **56.8** [50.1, 59.4] | **40.9** [37.3, 45.2] |

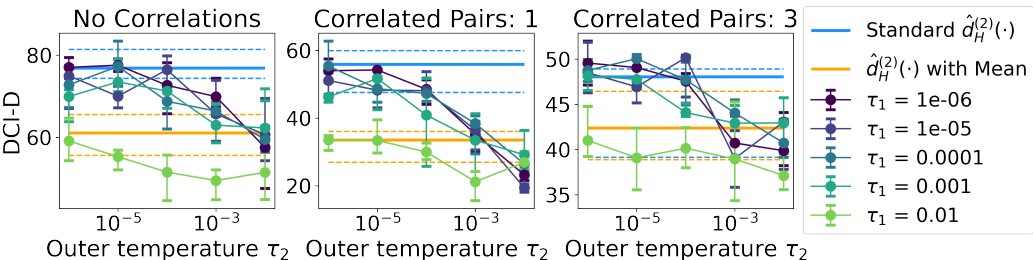

Figure 5: Results for our soft approximation to Eq. 5. Blue horizontal line denotes the default Eq. 5 objective, while orange denotes a replaced of the $\max$-operation with a mean. We find that generally, a convergence of the soft approach to our default hard variant performs best, with large choices in the outer temperature converging towards our mean approximation.

Here, the temperature $\tau_1$ controls the translation between putting more weight on the minimal distance (smaller $\tau_1$) versus a more uniform distribution (larger $\tau_1$), moving further away from the corresponding $\min$ operation.

A secondary $\tau_2$ then controls the transition between the $\max$-operation over our soft distances and a more uniform weighting over all (non-zero) soft distances, with the limit case $\tau_2 \to \infty$ approximating the simple mean over soft distances.

## B EVALUATION OF HAUSDORFF DISTANCE APPROXIMATION VARIANTS

As our utilized distance function $\hat{d}_H^{(2)}$ only approximates the Hausdorff distance to the factorized support, we now move to a variant study of other alternative distance measures as described above.

In particular, we investigate **(1)** a replacement of the $\max$-operation with a corresponding mean over support samples to address potential outliers better, **(2)** a probabilistic approximation to our $\min$-operation over $d(\mathbf{z}, \mathbf{z}')$ (see Eq. 11), **(3)** and a fully soft approximation to both $\max$ and $\min$ using a respective *Softmax* and *Softmin* formulation (A.4). **(4)** Finally, we also revisit the impact of exlicit scale regularization as introduced in §2.3.

**Method ablation.** Ablation studies across the default as well as two different training correlation settings can be found in Tab. 2, with each entry computed over at least 6 seeds, and a gridsearch over $\gamma \in \{1, 3, 10, 30, 100, 300, 1000, 3000, 10000\}$. Our results show that for optimization purposes, approximating the Hausdorff distances in a "sliced", pairwise fashion as suggested in Eq. 5 is noticeably better than subsampling from the incredibly high-dimensional factorized support, as instead of a single distance entry that is optimized for (after the $\max$-$\min$ selection), we have vari-

Table 3: Impact of the number of 2D approximations in $\hat{d}_H^{(2)}$. Our experiments reveal that the use of multiple 2D approximations to the full Hausdorff distances has notable merits up to a certain degree (change in disentanglement performance from e.g. $64\%$ to $75\%$ in the uncorrelated transfer setting). Each entry was chosen as the highest value in a gridsearch over $\gamma \in \{0.01, 0.1, 1, 10, 100\}$.

| Correlation ↓ | Num. Pairs: 1 | Num. Pairs: 2 | Num. Pairs: 5 | Num. Pairs: 15 | Num. Pairs: 25 | Num. Pairs: 35 | Num. Pairs: 45 |
|---|---|---|---|---|---|---|---|
| No Correlation | 64.0 [58.5, 68.7] | 66.3 [63.8, 72.6] | 67.9 [66.2, 69.5] | 76.6 [63.7, 79.5] | 74.1 [73.8, 77.6] | 75.5 [72.4, 79.9] | 74.5 [71.0, 76.0] |
| Correlated Pairs: 1 | 26.3 [21.0, 29.6] | 25.0 [23.9, 35.4] | 28.7 [24.5, 45.6] | 54.2 [44.2, 67.0] | 51.4 [42.5, 58.3] | 53.6 [48.3, 57.9] | 53.5 [51.0, 59.1] |
| Correlated Pairs: 3 | 40.9 [38.5, 44.3] | 44.3 [41.3, 48.0] | 46.8 [44.9, 48.9] | 46.5 [45.7, 47.3] | 48.7 [47.8, 50.2] | 48.8 [46.8, 52.0] | 49.2 [45.6, 51.8] |

Table 4: Impact of scale regularization (as detailed in §C) using VAE + $\hat{d}_H^{(2)}$ with $\gamma = 100$. For each setting, we perform a gridsearch over either the weight scale $\delta \in \{0, 1, 3, 10, 30, 100, 300, 1000, 3000, 10000\}$ or the L2 Regularization weight $\in \{10^{-6}, 10^{-5}, 10^{-4}, 10^{-3}, 10^{-2}, 10^{-1}\}$. Results show that scale regularization, while not necessarily detrimental, does not provide any consistent benefits.

| Setup ↓ Correlation → | No Correlation | Correlated Pairs: 1 | Correlated Pairs: 3 |
|---|---|---|---|
| Baseline | 74.1 [73.8, 77.6] | 51.4 [42.5, 58.3] | 48.7 [47.8, 50.2] |
| Variance | 72.4 [65.4, 76.1] | 54.4 [53.0, 55.6] | 50.1 [49.2, 53.3] |
| Min. Range | 75.0 [67.5, 78.5] | 50.6 [38.7, 55.2] | 51.5 [44.2, 53.3] |
| L2 Decoder | 73.4 [70.1, 78.2] | 56.3 [51.4, 60.0] | 52.9 [44.3, 52.2] |

ous latent subsets that incur a training gradient, and in two dimensions can cheaply compute the full factorized support.

Similarly, we also find that replacing the min-selection over latent entries with a probabilistic variant, as well as the max-selection over factorized support elements, offer no notable benefits. In particular the replacement of the outer max-operation can severely impact the disentanglement performance.

These insights are additionally supported when utilizing a soft variant (see Eq. A.4 in the appendix), which replaces both max and min operations with a respective Softmax and Softmin operation, each with respective temperatures $\tau_1$ and $\tau_2$. When utilizing this objective, we see that small temperature choices on both soft approximations are beneficial, and approximate the hard variant. Similarly, we find a consistent drop in performance when either one of these temperatures is reduced, with the soft performance converging towards the Mean variant when increasing the outer temperature $\tau_2$. Overall, we don't see any major benefits in a soft approximation, while also introducing two additional hyperparameters that would need to be optimized.

Finally, we ablate the key parameter for our Hausdorff distance approximation of choice, $\hat{d}_H^{(2)}$ (Eq. 5 - the number of pairs over which we compute a sliced 2D variant. Given a total latent dimensionality $k$, we are given $\binom{k}{2}$ usable combinations, which we can choose to subsample all the way down to a single pair of latent entries, which is what we do in Tab. 3. The results showcase that while a minimal number of latent pairs is crucial, not all combinations are needed, with diminishing returns for more pairs included. For practical purposes, we therefore choose 25 pairs as our default setting to strike a balance between performance and compute cost, which however can be easily increased if needed. On the latter note, we also highlight that while the addition of $\mathcal{L}_{HFS}$ does incur a higher epoch training time (60s for a standard VAE as used in Locatello et al. (2020b) on a NVIDIA Quadro GV100) than $\beta$-VAE (52s), it still compares favourably when compared to e.g. $\beta$-TCVAE (70s) or FactorVAE (96s). In addition, the impact on the training time diminishes when larger backbone networks are utilized.

## C REGULARIZING SCALE TO AVOID COLLAPSE

In the limit case, the standard Hausdorff matching problem is solved by collapsing all representations into a singular point. In addition to that, the actual scale of the latent entries directly impacts the distance scale. One can therefore provide additional regularization on top to ensure both a scale-invariant measure as well as work against a potential collapse, for example by enforcing a minimal

standard deviation $\rho$ on each factor[4]

$$\mathcal{L}_{\text{scale}} = \sum_{i=1}^{d} \max\left[0, 1 - \sqrt{\text{Var}\left[\mathbf{z}_{:,i}\right]}\right] \tag{14}$$

or simply enforcing a minimal range $[a, b]$ of the support:

$$\mathcal{L}_{\text{scale}} = \sum_{i=1}^{d} \max\left[0, b - \max(\mathbf{z}_{:,i})\right] + \max\left[0, \min(\mathbf{z}_{:,i}) - a\right] \tag{15}$$

In real setups however, we have found regularization of scale to not be necessary in the majority of cases, as the use of the additional autoencoding term alongside the Hausdorff Support Factorization is sufficient to avoid collapse (see part in §2.3 on collapse), as shown in Tab. 4. For Tab. 4, we also investigate what happens if we apply L2 regularization on the decoder.

In all cases, we perform a grid-search over an additional scale regularization weight parameter $\delta$ (with $\delta \in \{0, 1, 3, 10, 30, 100, 300, 1000, 3000, 10000\}$) or the L2 Regularization weight ($\in \{10^{-6}, 10^{-5}, 10^{-4}, 10^{-3}, 10^{-2}, 10^{-1}\}$). Our results show no improvements that are both significant and consistent across correlation settings. And while these regularization may become relevant for future variants and extensions, in this work we choose to forego a scale regularizer with the benefits of having one less hyperparameter to tune for.

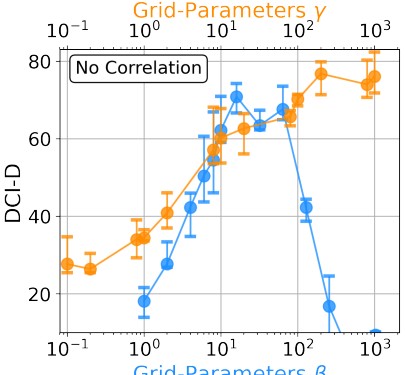

Figure 6: Robustness to factorized support weighting $\gamma$ - much less detrimental to overall training dynamics.

## D   HYPERPARAMETER EVALUATION

To understand to what extent the factorization of support parameter $\gamma$ impacts the learning and performance of the model, we also compare grid searches over $\gamma$ and the standard $\beta$-VAE prior matching weight $\beta$. The results in Fig. 6 indicate that a factorization of support is much less dependent on the exact choice of weighting $\gamma$ as opposed to the standard KL-Divergence to the normal prior used in $\beta$-VAE frameworks (notice the logarithmic value grid). This stands to reason, as a factorization of the support instead of distributions is both a more realistic property as well as a much weaker constraint on the overall training dynamics.

## E   SAMPLE RECONSTRUCTIONS AND QUALITATIVE EVALUATION

We also provide some qualitative impression of the impact an explicit factorization has on the overall disentanglement across different correlations. In particular, Figure 7 visuals latent traversals both for the $\beta$-VAE baseline (top)as well as the HFS-augmented variant (bottom) for the latent entry most expressive for the first mentioned latent entry in ("Correlations addressed"). To generate these figures, we select the best performing seed for each setup, and report the respective DCI-D score within each subplot. As can be seen, beyond the increase in maximally achievable DCI-D, an explicit factorization of the support helps the disentangling method separate factors it initially struggled with - both when correlations exists in the training data as well as for generally failure modes when the $\beta$-VAE fails to fully disentangle in the uncorrelated setting.

---

[4]Though this enforces an implicit assumption on the density within each latent factor.

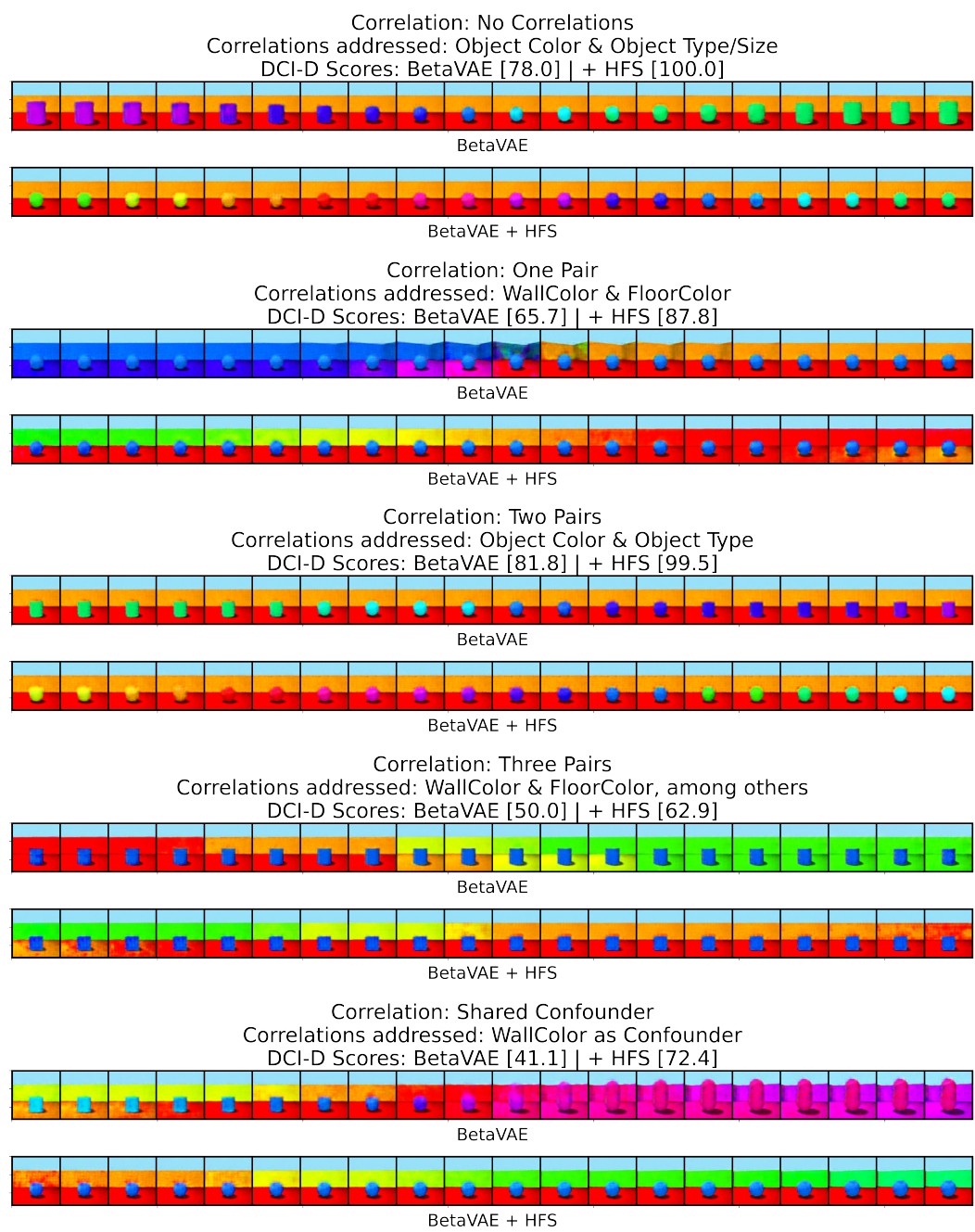

Figure 7: Sample traversals for the Shapes3D benchmark Kim & Mnih (2018) in latent space for latent entry most closely associated with various ground truth factors of variations across different correlation shifts. In all cases, the best seed (out of 10) was selected to perform these qualitative studies. Each image also reports the associated overall DCI-D score of each respective best seed for $\beta$-VAE and $\beta$-VAE + HFS.

| Method | No Corr. | Pair: 1 [V1, σ = 0.1] | Pair: 1 [V2, σ = 0.1] | Pair: 1 [V3, σ = 0.1] | Pair: 1 [V4, σ = 0.1] | Pair: 1 [V4-inv, σ = 0.1] | Pairs: 2 [V1, σ = 0.4] | Pairs: 2 [V2, σ = 0.4] | Pairs: 2 [V1, σ = 0.1] | Pairs: 2 [V2, σ = 0.1] | Pairs: 2 [V3, σ = 0.1] | Pairs: 2 [V3-inv, σ = 0.1] | Pairs: 3 [V1, σ = 0.1] | Pairs: 3 [V2, σ = 0.1] | Conf. [V1, σ = 0.2] | Conf. [V2, σ = 0.2] |
|---|---|---|---|---|---|---|---|---|---|---|---|---|---|---|---|---|
| β-VAE | 70.7 [65.2, 75.4] | 55.9 [53.4, 59.0] | 71.6 [60.9, 72.5] | 66.1 [62.8, 67.0] | 64.8 [62.0, 66.4] | 64.5 [62.6, 66.1] | 68.5 [56.8, 76.9] | 71.5 [68.1, 75.1] | 58.7 [57.1, 62.9] | 55.6 [45.8, 57.2] | 60.2 [54.8, 61.6] | 55.5 [52.8, 56.2] | 37.1 [32.3, 38.9] | 40.0 [33.8, 47.5] | 46.8 [42.0, 50.7] | 38.0 [35.0, 38.6] |
| HFS | 78.3 [75.1, 83.6] | 56.4 [46.0, 60.2] | 77.8 [74.4, 78.8] | 68.1 [61.2, 69.4] | 71.0 [65.8, 73.6] | 69.6 [68.2, 71.0] | 73.9 [65.5, 80.0] | 76.7 [69.7, 81.2] | 62.6 [61.2, 64.3] | 56.0 [41.0, 57.4] | 64.4 [62.3, 65.3] | 61.5 [57.5, 62.6] | 47.5 [38.0, 49.1] | 41.1 [31.7, 42.7] | 51.3 [46.2, 52.5] | 46.2 [36.7, 47.5] |
| β-VAE + HFS | **91.2** [75.8, 100.0] | **67.3** [39.5, 72.3] | **80.9** [76.4, 81.4] | **76.1** [72.2, 79.5] | **75.8** [67.1, 78.9] | **74.0** [71.8, 78.9] | **89.1** [78.2, 98.5] | **83.3** [79.0, 86.5] | **65.5** [62.0, 66.6] | **67.6** [62.4, 69.3] | **65.8** [63.2, 68.4] | **63.8** [57.7, 65.4] | **47.9** [44.0, 50.8] | **52.0** [48.7, 54.5] | **59.7** [58.6, 62.1] | **63.5** [61.2, 65.5] |
| β-TCVAE | 77.1 [76.6, 78.3] | 62.0 [56.6, 64.5] | 71.1 [65.5, 72.5] | 69.8 [67.3, 70.8] | 63.2 [61.7, 66.6] | 66.4 [65.1, 66.7] | 75.8 [73.0, 79.1] | 75.2 [69.0, 75.9] | 63.7 [62.9, 70.0] | 63.8 [59.1, 65.1] | 60.1 [53.6, 62.7] | 54.3 [50.8, 54.8] | 47.3 [36.7, 50.0] | 58.1 [56.2, 61.1] | 55.9 [52.7, 59.9] | 49.9 [45.8, 55.9] |
| FactorVAE | 66.1 [51.2, 69.1] | 52.2 [44.7, 54.8] | 70.8 [70.5, 71.2] | 65.9 [64.2, 67.8] | 64.6 [63.7, 64.8] | 63.5 [61.9, 64.4] | 70.2 [64.8, 75.0] | 71.2 [63.0, 77.7] | 62.0 [60.7, 64.5] | 57.2 [55.9, 62.0] | 60.3 [52.9, 60.9] | 56.8 [53.3, 57.6] | 46.8 [40.8, 49.0] | 40.2 [35.2, 44.6] | 39.2 [34.2, 47.8] | 31.6 [27.9, 35.1] |
| AnnealedVAE | 62.2 [60.7, 63.2] | 39.6 [29.6, 41.6] | 57.2 [49.5, 59.3] | 56.3 [53.0, 57.2] | 58.0 [49.9, 60.7] | 58.9 [37.6, 61.2] | 60.4 [59.3, 64.9] | 48.5 [39.7, 49.1] | 50.9 [46.0, 52.9] | 31.6 [26.9, 34.1] | 50.7 [48.5, 53.4] | 51.3 [49.5, 52.3] | 33.6 [31.0, 38.0] | 30.2 [27.1, 30.8] | 26.2 [22.0, 26.8] | 23.0 [20.1, 25.9] |

(a) Shapes3D

| Method | No Corr. | Pair: 1 [V1, σ = 0.1] | Pair: 1 [V2, σ = 0.1] | Pair: 1 [V3, σ = 0.1] | Pairs: 2 [V1, σ = 0.4] | Pairs: 2 [V2, σ = 0.4] | Pairs: 2 [V1, σ = 0.1] | Pairs: 2 [V2, σ = 0.1] | Pairs: 3 [V1, σ = 0.1] | Pairs: 3 [V2, σ = 0.1] | Conf. [V1, σ = 0.2] | Conf. [V2, σ = 0.2] |
|---|---|---|---|---|---|---|---|---|---|---|---|---|
| β-VAE | 25.6 [24.7, 26.1] | 15.7 [13.9, 17.0] | 20.5 [17.7, 20.9] | 23.5 [22.5, 24.4] | 23.6 [21.3, 24.7] | 24.8 [24.5, 25.9] | 21.2 [19.5, 21.7] | 23.6 [22.6, 24.3] | 11.6 [11.1, 11.7] | 11.1 [10.9, 11.3] | 15.1 [14.5, 15.8] | 11.8 [10.0, 12.7] |
| HFS | **32.8** [30.0, 34.3] | **20.7** [19.5, 21.2] | **28.4** [26.5, 29.5] | **26.9** [24.7, 28.0] | **30.1** [29.7, 31.0] | **30.2** [29.5, 30.5] | 25.6 [24.0, 26.2] | **28.0** [27.4, 28.2] | **14.3** [13.1, 14.8] | **19.0** [17.8, 19.3] | **18.9** [14.4, 19.2] | **16.1** [15.0, 16.6] |
| β-VAE + HFS | **32.8** [30.0, 34.3] | **20.7** [19.5, 21.2] | **28.4** [26.5, 29.5] | **26.9** [24.7, 28.0] | **30.1** [29.7, 31.0] | **30.2** [29.5, 30.5] | 25.6 [24.0, 26.2] | **28.0** [27.4, 28.2] | **14.3** [13.1, 14.8] | **19.0** [17.8, 19.3] | **18.9** [14.4, 19.2] | **16.1** [15.0, 16.6] |
| β-TCVAE | 26.6 [26.0, 27.4] | **21.6** [20.4, 23.8] | 20.7 [20.4, 21.3] | 23.7 [23.5, 24.2] | 25.6 [25.1, 25.9] | 25.6 [25.4, 26.2] | 21.6 [19.9, 23.2] | 23.3 [21.9, 23.8] | 11.4 [10.3, 12.6] | 16.5 [15.4, 18.2] | 16.7 [16.0, 17.1] | 14.2 [13.4, 15.4] |
| FactorVAE | 26.0 [24.8, 27.5] | 20.1 [15.5, 22.7] | 21.9 [20.1, 23.9] | 24.6 [23.8, 26.2] | 25.0 [24.0, 25.9] | 27.8 [27.2, 29.2] | 21.9 [18.6, 24.0] | 21.6 [17.6, 24.4] | 10.9 [10.7, 11.9] | 15.4 [14.8, 16.4] | 15.5 [15.0, 16.5] | 13.6 [12.8, 13.9] |
| AnnealedVAE | 11.8 [10.8, 12.4] | 10.8 [9.6, 11.9] | 11.7 [10.4, 12.9] | 10.6 [10.3, 11.9] | 12.9 [11.0, 14.8] | 11.8 [11.6, 12.1] | 10.8 [9.8, 11.6] | 12.5 [10.1, 13.5] | 11.6 [10.6, 12.2] | 10.1 [9.8, 11.0] | 13.3 [11.4, 13.8] | 13.4 [12.8, 13.9] |

(b) MPI3D

| Method | No Corr. | Pair: 1 [V1, σ = 0.1] | Pair: 1 [V2, σ = 0.1] | Pair: 1 [V3, σ = 0.1] | Pairs: 2 [V1, σ = 0.4] | Pairs: 2 [V2, σ = 0.1] | Pairs: 2 [V1, σ = 0.4] | Pairs: 2 [V2, σ = 0.1] | Conf. [V1, σ = 0.2] | Conf. [V2, σ = 0.2] |
|---|---|---|---|---|---|---|---|---|---|---|
| β-VAE | 32.2 [25.3, 37.9] | 17.9 [10.4, 23.0] | 9.5 [7.9, 10.3] | 13.5 [9.8, 16.1] | 20.5 [18.7, 27.6] | 7.5 [6.7, 8.3] | 24.8 [11.0, 27.6] | 10.0 [6.8, 12.3] | 14.0 [10.4, 18.5] | 11.4 [9.9, 13.9] |
| HFS | 34.9 [27.4, 36.0] | 22.7 [14.4, 25.6] | 13.6 [7.6, 16.7] | 23.3 [13.8, 28.4] | 24.1 [13.3, 30.3] | 11.9 [9.7, 13.8] | 24.3 [21.7, 25.6] | 11.1 [9.8, 11.3] | 15.8 [5.4, 18.9] | 15.1 [11.0, 16.0] |
| β-VAE + HFS | **49.9** [30.0, 50.4] | 32.9 [21.3, 39.0] | **19.7** [17.0, 21.1] | **37.5** [25.9, 39.1] | **38.2** [21.9, 41.7] | **17.3** [6.0, 19.6] | 32.7 [23.1, 33.0] | 14.6 [14.3, 14.9] | 22.1 [17.7, 24.4] | **15.8** [12.3, 16.7] |
| β-TCVAE | 35.3 [29.5, 38.8] | **35.1** [32.3, 38.4] | **24.0** [23.6, 24.4] | 25.0 [19.0, 30.9] | 30.2 [27.9, 42.2] | 11.4 [7.6, 13.6] | **33.0** [24.3, 37.8] | **20.7** [16.5, 21.9] | **29.4** [28.7, 30.9] | **20.9** [17.5, 23.6] |
| FactorVAE | 25.7 [20.9, 30.9] | 22.6 [20.8, 25.4] | 15.1 [11.9, 16.3] | 21.2 [19.2, 24.6] | 21.2 [13.5, 22.9] | 13.4 [12.4, 15.0] | 23.4 [22.3, 26.1] | 13.0 [6.5, 14.4] | 18.3 [17.5, 19.2] | 14.7 [13.5, 15.3] |
| AnnealedVAE | 39.4 [38.7, 40.0] | **40.8** [39.4, 41.3] | 14.8 [14.3, 15.9] | 29.0 [26.6, 29.9] | 30.3 [28.3, 31.4] | 8.5 [6.9, 10.3] | 28.3 [27.6, 28.5] | 10.1 [9.3, 10.6] | 19.1 [18.2, 19.2] | 14.3 [14.1, 14.5] |

(c) DSprites

Table 5: Full table for Tab. 1 with detailed and extended correlation settings. To understand the exact factors correlated, please check the associated pairings from §H.

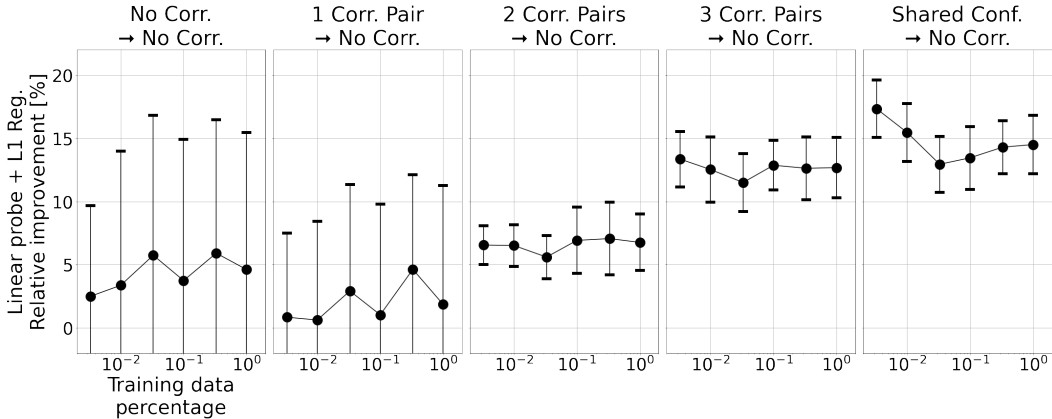

Figure 8: This figure shows adaptation behaviour across different amounts of test data for a L1-optimal linear probe (i.e. for each seed and entry, we selected the optimal L1-regularization values). Reported values show relative improvement in average ground truth factor classification performance of β-VAE + **HFS** versus standard β-VAE. As can be seen, the increased disentanglement through an explicitly factorized support gives expected improvements increasing with the severity of training correlations encountered.

## F DETAILED FIGURES AND TABLES

In this section, we provide detailed variants of figures and tables utilised in the main paper.

### F.1 ADDITIONAL DISENTANGLEMENT RESULTS

For Tab. 1 studying the impact of HFS both as a standalone objective and as a regularizer on disentanglement of test data across varying degrees of training correlations, we include a more detailed variant highlighting the exact splits utilised in Tab. 5, as well as additional correlation settings.

### F.2 FURTHER ADAPTATIONS

Extending our adaptation experiments done in §4.3, we also investigate the average classification performance of ground truth factors of a weaker, L1-regularized linear probe in Fig. 8. Similar to

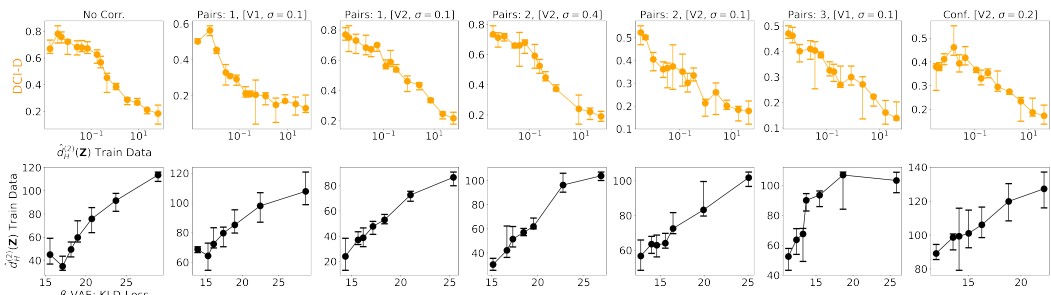

Figure 9: This is the full figure for Fig. 4, showcasing that a factorization of the support on the training data is consistently linked to improved downstream disentanglement (*top*), and that a minimization of the standard $\beta$-VAE KLD-objective for a factorial distribution implicitly minimizes for a factorized support across settings.

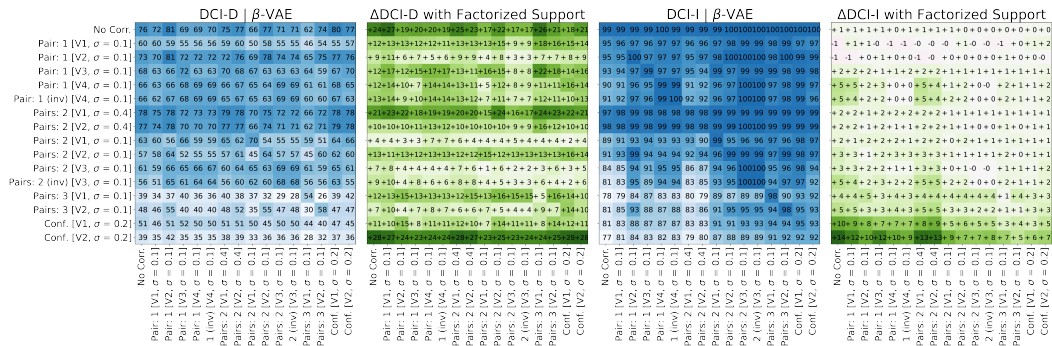

Figure 10: This is the detailed correlation shift transfer grid utilized in Fig. 2, indicating the exact correlation settings used for training and test data.

our transfer results, we again find that the increased disentanglement through an explicitly factorized support gives expected improvements which increasing with the severity of training correlations.

## F.3 FURTHER PROGRESSIONS

Fig. 9 provides visualization of the relations between training support factorization and disentanglement performance on test data (top), as well the KL-Divergence loss in the standard $\beta$-VAE for more training correlation settings as shown in the main paper figure 4(orange and black graphs), with insights transferring from the main paper.

## F.4 DETAILED CORRELATION SHIFT TRANSFER GRID

For replicability, we provide a copy of Fig. 2 with the exact utilised correlation settings in Fig. 10.

## F.5 DETAILED METRIC GRID

Finally, we also provide a more detailed copy of the metric transfer grid in Fig. 4(right, blue) with the exact correlation settings investigated.

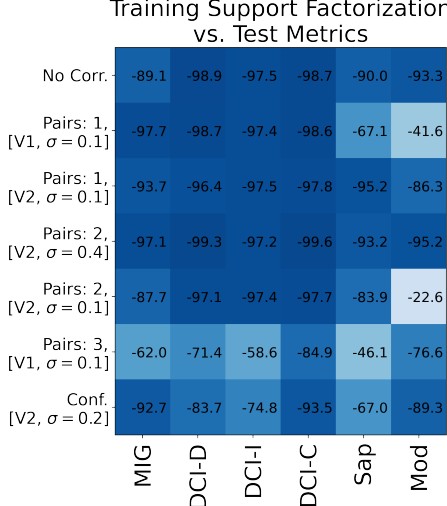

Figure 11: Correlation of Hausdorff distance to factorized support on the training data to various disentanglement metrics (in particular DCI and MIG) across correlation shifts. We find factorization of supports on the training data to strongly relate to downstream disentanglement even when experiencing strong correlation during training.

## G DISENTANGLEMENT METRICS

In this section, we will provide a brief introduction into various disentanglement metrics, with particular emphasis on the DCI-D metric (Eastwood & Williams, 2018) used as our leading measure of disentanglement.

### G.1 DCI-D AND DCI-I

DCI-**D**isentanglement was introduced in Eastwood & Williams (2018) as part of a three-property description of learned representation spaces, alongside **C**ompleteness and **I**nformativeness. In this work, we primarily utilize DCI-D as a measure of disentanglement, and DCI-I as a measure of generalization performance. In particular, each submetric utilizes multiple classification models (e.g. logistic regressor (Eastwood & Williams, 2018) or a boosted decision tree (Locatello et al., 2019b)), which are trained to predict each underlying ground-truth factors from representations extracted from the dataset of interest, respectively. DCI-I is then simply computed as the average prediction error (on a test-split). To compute DCI-D, for each ground-truth factor and consequently each prediction model, predictive importance scores for each dimension of the representation space are extracted from the classification model, given as $R \in \mathbb{R}^{d \times k}$ with representation dimensionality $d$ and number of factors $k$. For each row, the entropy value is then computed and subtracted from 1 - being high if a dimension is predictive for only one factor, and low if it is used to predict multiple factors. Finally, each entropy score is weighted with the relative overall importance of the respective dimension to predict any of the ground-truth factors, giving

$$\text{DCI-D} = \sum_i^d (1 - H(\text{Norm}(R_{i,:}))) \sum_j^k \frac{R_{i,j}}{\sum_{i^*} \sum_{j^*} R_{i^*,j^*}}$$

### G.2 MUTUAL INFORMATION GAP (MIG)

The Mutual Information Gap (MIG) was introduced in Kim & Mnih (2018) to measure the mutual information difference of the two representation entries that have the highest mutual information with a respective ground-truth factor normalized by the respective entropy, which is then averaged for all ground-truth factors. For our work, we follow the particular formulation and implementation introduced in Locatello et al. (2019b), by taking the mean representations produced by the encoder

network, and estimating a discrete mutual information score, such that the overall MIG can be computed as

$$\text{MIG} = \frac{1}{k} \sum_{i=1}^{k} \frac{I(\tilde{z}_{\tilde{m}(k,1)}, z_k) - I(\tilde{z}_{\tilde{m}(k,2)}, z_k)}{H(z_k)}$$

where $k$ denotes the number of ground-truth factors of variation $z_k$, $H(z_k)$ the respective entropy of $z_k$, $\tilde{z}_i$ the $i$-th entry of the generated latent space, and $\tilde{m}(k, n)$ a function that returns the representation index with the $n$-th highest mutual information to ground-truth factor $k$. To compute the discrete mutual information, we get distribution estimates by binning representation values for each dimensions across 20 bins, doing so over 10000 samples.

### G.3 MODULARITY

Ridgeway & Mozer (2018) introduce the notions of *Modularity* and *Expressiveness* as key components of a disentangled representation - with the former evaluating whether each representation dimension depends on at most a single ground-truth factor of variation, and the latter the predictiveness of the overall representation to predict ground-truth factor values. Similarly to Locatello et al. (2019b), we mainly focus on the property of *Modularity*, which Ridgeway & Mozer (2018) define for a $d$-dimensional representation space with $k$ ground-truth factors as

$$\text{Modularity} = \frac{1}{d} \sum_{i}^{d} \frac{\sum_{j}(m_{i,j} \cdot \mathbb{I}_{j=\text{argmax}_g m_{i,g}})^2}{(\max_g m_{i,g})^2 (k-1)} \tag{16}$$

which, per latent dimension $i$ measures the average normalized squared mutual information scores between the factors that do not share the highest mutual information with the latent entry $i$. Here, $m_{i,j}$ denotes the discretized mutual information between latent entry $i$ and factor $j$ similar to our implementation of the Mutual Information Gap and Locatello et al. (2019b), where we utilize a discretized approximation by binning each latent entry into 20 bins over 10000 samples to compute the discretized mutual information scores.

### G.4 SAP SCORE

The *Separated Attribute Predictability* (SAP) score was introduced in Kumar et al. (2018) as another disentanglement measure, in which the authors suggest to train a linear regressor (in the case of Locatello et al. (2019b)a linear SVM with C = 0.01 and again 10000 training samples and 5000 test points) to predict each ground-truth factor from each dimension of the learned representation space, and then taking the average difference in prediction errors between the two most predictive latent entries for each respective ground-truth factor.

### G.5 BETA- AND FACTORVAE SCORES

The FactorVAE Score (Kim & Mnih, 2018) is an extension of the BetaVAE Score introduced in Higgins et al. (2017). In both cases, a ground-truth factor of variation is fixed, and two sets of observations are then sampled. The BetaVAE score then measures disentanglement as the classification accuracy of a linear classifier to predict the index of the fixed factor based on the average absolute differences between set pairs. In Locatello et al. (2019b), this process is repeated 10000 times to train a logistic regressor, and evaluated on 5000 test pairs. The FactorVAE score improves on this metric through the use of a majority vote classifier that instead predicts based on the index of the representation entry with least variance.

| Method | Parameter | Values |
|:---:|:---:|:---:|
| $\beta$-VAE | $\beta$ | [1, 2, 3, 4, 6, 8, 10, 12, 16] |
| $\beta$-TCVAE | $\beta$ | [1, 2, 3, 4, 6, 8, 10, 12, 16] |
| AnnealedVAE | $c_{\max}$ | [2, 5, 10, 25, 50, 75, 100, 150] |
| FactorVAE | $\beta$ | [2, 5, 10, 25, 50, 75, 100, 150] |
| HFS | $\gamma$ | [20, 40, 80, 100, 200, 400, 800, 1000, 2000, 4000] |
| $\beta$-VAE + HFS | $\gamma$ | [30, 60, 100, 300, 600, 1000, 3000, 6000] |

Table 6: Hyperparameter grid searches for different baseline methods as well as our factorized support objective.

## H  FURTHER EXPERIMENTAL DETAILS

**Study design.** We implement all our experiments using the PyTorch framework Paszke et al. (2019). For exact and fair comparability, we re-implement all baseline methods based on Tensorflow implementation provided through Locatello et al. (2019b) (https://github.com/google-research/disentanglement_lib), as well as the following public repositories: https://github.com/YannDubs/disentangling-vae, https://github.com/nmichlo/disent/blob/main, https://github.com/ubisoft/ubisoft-laforge-disentanglement-metrics/blob/main/src/metrics/dci.py and https://github.com/AntixK/PyTorch-VAE.

For the implementation of the disentanglement metrics, we follow the implementation used in Locatello et al. (2019b), which for the computation of the DCI metrics leverages a gradient boosted tree from the scikit-learn package. The VAE architecture used throughout our experiments follows the one utilized in Locatello et al. (2020b), which leverages the following architecture, assuming input image sizes of $64 \times 64 \times n_c$ with $n_c$ the number of input channels, usually 3, and a latent dimensionality of 10:

- **Encoder**: [conv(32, $4 \times 4$, stride 2) + ReLU] $\times$ 2, [conv(64, $4 \times 4$, stride 2) + ReLU] $\times$ 2, MLP(256), MLP($2 \times 10$)
- **Decoder**: MLP(256), [upconv(64, $4 \times 4$, stride 2) + ReLU] $\times$ 2, [upconv(32, $4 \times 4$, stride 2) + ReLU], [upconv($n_c$, $4 \times 4$, stride 2) + ReLU]

The training details are as follows:

- Optimization: Batchsize = 64, Optimizer = Adam ($\beta_1 = 0.9, \beta_2 = 0.999, \epsilon = 10^{-8}$), Learning rate = $10^{-4}$.
- Training: Decoder distribution = Bernoulli, Training steps = 300000

Note that FactorVAE Kim & Mnih (2018) introduces a separately trained discriminator, we we again utilize the setting described in Locatello et al. (2020b):

- Architecture: [MLP(1000), leakyReLU] x 6, MLP(2)
- Optimization: Batchsize = 64, Optimizer = Adam ($\beta_1 = 0.5, \beta_2 = 0.9, \epsilon = 10^{-8}$)

Finally, we provide the hyperparameter gridsearches performed for every baseline method which mostly follow Locatello et al. (2020b), as well as for HFS (though for some ablation studies more coarse-grained grids very utilised) and $\beta$-VAE + HFS: Note that for AnnealedVAE, we also leverage an iteration threshold of $10^5$ and $\gamma = 10^3$.

### H.1  CORRELATION SETTINGS

We now provide more detailed information regarding the specific abbrevations used throughout the main text and for the following appendix to denote various correlation setups during training. We note that to introduce multiple correlated factors pairs, we simply multiply respective $p(c_i, c_j)$ entries.

For Shapes3D (Kim & Mnih, 2018), we introduce the following correlations:

- `No Corr.`: No Correlation during training. This constitutes the default evaluation setting.
- `Pair: 1 [V1]`: `floorCol` and `wallCol`.
- `Pair: 1 [V2]`: `objType` and `objSize`.
- `Pair: 1 [V3]`: `objType` and `wallCol`.
- `Pair: 1 [V4]`: `objType` and `objCol`.
- `Pair: 1 [inv, V4]`: `objType` and `objCol`, but inverse correlation.
- `Pairs: 2 [V1]`: `objSize` and `floorCol` as well as `objType` and `wallCol`.
- `Pairs: 2 [V2]`: `objSize` and `objType` as well as `floorCol` and `wallCol`.
- `Pairs: 2 [V3]`: `objType` and `objCol` as well as `objType` and `objSize`.
- `Pairs: 2 [inv, V3]`: `objType` and `objCol` as well as `objType` and `objSize`, but with inverse correlation.
- `Pairs: 3 [V1]`: `objSize` and `objAzimuth` as well as `objType` and `wallCol`, and `objCol` and `floorCol`.
- `Pairs: 3 [V2]`: `objCol` and `objAzimuth` as well as `objType` and `objSize`, and `wallCol` and `floorCol`.
- `Shared Conf. [V1]`: We correlate (confound) `objType` against all other factors.
- `Shared Conf. [V2]`: We correlate (confound) `wallCol` against all other factors.

For MPI3D (Gondal et al., 2019), we introduce the following correlations:

- `No Corr.`: No Correlation during training. This constitutes the default evaluation setting.
- `Pair: 1 [V1]`: `cameraHeight` and `backgroundCol`.
- `Pair: 1 [V2]`: `objCol` and `objSize`.
- `Pair: 1 [V3]`: `posX` and `posY`.
- `Pairs: 2 [V1]`: `objCol` and `objShape` as well as `posX` and `posY`.
- `Pairs: 2 [V2]`: `objCol` and `posX` as well as `objShape` and `posY`.
- `Pairs: 3 [V1]`: `objCol` and `backgroundCol` as well as `cameraHeight` and `posX`, and `objShape` and `posY`.
- `Pairs: 3 [V2]`: `objCol` and `posX` as well as `objShape` and `posY`, and `backgroundCol` and `cameraHeight`.
- `Shared Conf. [V1]`: We correlate (confound) `objShape` against all other factors.
- `Shared Conf. [V2]`: We correlate (confound) `posX` against all other factors.

For DSprites (Higgins et al., 2017), we introduce the following correlations:

- `No Corr.`: No Correlation during training. This constitutes the default evaluation setting.
- `Pair: 1 [V1]`: `shape` and `scale`.
- `Pair: 1 [V2]`: `posX` and `posY`.
- `Pair: 1 [V3]`: `shape` and `posY`.
- `Pairs: 2 [V1]`: `shape` and `scale` as well as `posX` and `posY`.
- `Pairs: 2 [V2]`: `shape` and `posX` as well as `scale` and `posY`.
- `Shared Conf. [V1]`: We correlate (confound) `shape` against all other factors.
- `Shared Conf. [V2]`: We correlate (confound) `posX` against all other factors.

## I  PSEUDOCODE

Finally, we provide a PyTorch-style pseudocode to quickly re-implement and apply the factorization objective following Eq. 5.

```python
# Inputs:
# * Batch of latents <z> [bs x dim]
# * Number of latent pairs to use for approximation <n_pairs_to_use>.

import itertools as it
import numpy as np
import torch

# Get available latent pairs.
pairs = np.array(list(it.combinations(range(dim), 2)))
n_pairs = len(pairs)
pairs = pairs[np.random.choice(n_pairs, n_pairs_to_use, replace=False)]

# Subsample batch <z> [bs x latent_dim] into <s_z> [bs x num_latent_pairs
      x 2]
s_z = z[..., pairs]

# ixs_a = [0, ..., bs-1, 0, 1, ...., bs-1]
ref_range = torch.arange(len(z), device=z.device)
ixs_a = torch.tile(ref_range, dims=(len(z),))
# ixs_b = [0, 0, 0, ..., 1, 1, ..., bs-1]
ixs_b = torch.repeat_interleave(ref_range, len(z))

# Aggregate factorized support:
#    For every latent pair, we select all possible batch pairwise
#    combinations, giving our factorized support <fact_z>:
#    dim(fact_z) = bs **2 x num_latent_pairs x 2
fact_z = torch.cat([s_z[ixs_a, :, 0:1], s_z[ixs_b, :, 1:2]], dim=-1)

# Compute distance between factorized support and 2D batch embeddings:
# dim(dists) = bs ** 2 x bs x num_pairs
dists = ((fact_z.unsqueeze(1) - s_z.unsqueeze(0)) ** 2).sum(-1)

# Compute Hausdorff distance for each pair, then sum up each pair
      contribution.
hfs_distance = dists.min(1)[0].max(0)[0].sum()
```

Listing 1: Sample PyTorch Implementation of $\hat{d}_H^{(2)}$ (Eq. 5)

