# OpenReview forum: "Disentanglement of Correlated Factors via Hausdorff Factorized Support"
_ICLR.cc/2023/Conference — ICLR 2023 poster_

### Official Review · Reviewer_e6Ls · 2022-10-19

**Confidence:** 4
**Correctness:** 3
**Technical Novelty And Significance:** 3
**Empirical Novelty And Significance:** 3
**Recommendation:** 8

**Clarity, Quality, Novelty And Reproducibility:**

Clarity: The paper is well written and clear.
Quality: The main claims seem to be supported, the experimental results and ablation are well done.
Novelty: I am not aware of previous approaches to unsupervised disentanglement based on support factoriization.
Reproducibility: Code was provided.

**Strength And Weaknesses:**

+ Removing the assumption of independence between factors of variation in disentanglement models is of high importance.
+ The approach of independent supports may be a promising direction for tackling this task and appears novel.
+ The formulation of the loss term seems reasonable (but does not seem to be the entire story)
+ Experimental results do seem to confirm the claims of the paper (but note weaknesses)

- The loss term is only able to detect second order correlations, which is not quite the same as ensuring factorized support. While the authors explained why this did it, it would have been interesting to understand the performance loss due to this relaxation.
- The experiments are somewhat self-serving in the sense the correlation is linear between pairs of terms, which is also what the loss term is optimising. How would it deal with higher order correlations?
- The loss term, combined with reconstruction was insufficient and required the KL divergence with the normal distribution, which encourages no correlation. While the authors tried to address it in the paper, I wish that they justify this finding in a clearer way.
- The datasets useed are somewhat easy. How does this method performance on the more challenging SmallNorb?

**Summary Of The Paper:**

The paper tackles a key problem in disentanglement - the independence assumption made by most methods does not hold in practice. Instead, the paper suggests requiring independent supports between factors which can be satisfied even when correlation exists. The paper proposes to measure support independence by the empirical Hausdorff distance, but find that doing so is infeasible. Instead, it measures the hausdorff distance on slices of the representations (specifically pairs of factors). While not powerful enough on its own, the authors show the Hausdorff loss term is very helpful when combined with existing approaches. Benefits are shown, particularly when strong correlations exist in the data. Good results are also shown for OOD generalization.

**Summary Of The Review:**

Overall, I find the idea interesting and the experimental evidence positive. Please address the weaknesses above. I am leaning towards acceptance.

#######################

Post-rebuttal: The authors mostly addressed my concerns, I keep my rating and recommend acceptance.

---

> ### Author Response · Authors · 2022-11-09
> **RE: Official Review of Paper4621 by Reviewer e6Ls**
>
> We would firstly like to thank the reviewer for the detailed review and feedback, and address each noted question separately.
>
> ___The loss term is only able to detect second order correlations, which is not quite the same as ensuring factorized support. While the authors explained why this did it, it would have been interesting to understand the performance loss due to this relaxation.___
>
> The reviewer correctly points out that our additional relaxation means that we focus on factor pairs instead of the full support. However, we do so in a “sliced” fashion to approximate full support factorization, meaning that we require factorization for any factor pairs.
> And indeed, when we compare our pairwise relaxation to a factorization approach on the full support using subsampling (App. Tab. 2), we perform much better, and much more cost-efficient (with the highest subsampling count requiring over twice the compute time and memory).
> As such, from a practical point-of-view, our pairwise relaxation actually notably benefits the disentanglement performance.
>
> ___The experiments are somewhat self-serving in the sense the correlation is linear between pairs of terms, which is also what the loss term is optimising. How would it deal with higher order correlations?___
>
> While it is true that our relaxation only optimizes for factor pairs, it does so for any factor pair, with our experiments (App. Tab. 2) showing that this sliced approach both performs and scales better than the fully-subsampled variant.
> And regarding the correlations, we use the established correlation formalism introduced in Traeuble et al. 2021, but make it even harder by including both harder correlation degrees as well as multiple correlated pairs and shared confounders, which even introduce implicit correlations between factor pairs. In all cases, support factorization offers significant improvements in disentanglement.
> Finally, we do agree that investigating higher order correlations would have been interesting as well, but which we leave to future work to investigate in more detail, as this would have gone beyond the scope of this work which is to provide a proof-of-concept method to allow for disentangled representation learning under correlations and correlations shift, and would have also increased the compute required for the correlation shift experiments too much.
> We do want to note that a factorization of the support should work regardless of the specific type of correlation existent in the data!
>
> ___The loss term, combined with reconstruction was insufficient and required the KL divergence with the normal distribution, which encourages no correlation. While the authors tried to address it in the paper, I wish that they justify this finding in a clearer way.___
>
> We kindly disagree with the statement that HFS (without KL-divergence) was insufficient - looking at Tab. 2, we find that in the standard, uncorrelated setup, we generally outperform or match the other methods, with the performance gap __only becoming more evident__ as the specific independence-assumption-based regularization is applied on top, with our leading hypothesis for this explained in our reply to Reviewer 1:
> _We find that when we use HFS as a regularizer on top of these independence-assuming approaches, we get even higher disentanglement performance, which we attribute to the fact that while we do have correlations in most of our settings, we generally have various other factor pairs for which the independence assumption remains valid (as only some pairs of factors are correlated), and for which allowing for a small degree of factor independence benefits overall disentanglement performance, which supports HFS being a more relaxed constraint, and factorized support a generally beneficial property to account for. Conversely, as our benchmarks always have some factor pairs that remain (mostly) independent, the independence assumption does not generally become a breaking property, and can work well in conjunction with HFS._
>
> ___The datasets useed are somewhat easy. How does this method performance on the more challenging SmallNorb?___
>
> We use standard benchmarks, of which SmallNORB is just another even older one, with for example MPI3D being a much more involved and complicated benchmark, on which we also outperform other methods.
> As an example, the default SmallNORB dataset only contains 24300 images over five ground-truth factors of variation of clearly background-separated toys (which are even less images and factors than the Shapes3D dataset), wheres the MPI3D dataset contains over a million images over seven factors of variation from a realistic robot arm simulation.
> As such, from the standpoint of investigating benchmarks with controlled factors of variation, we believe to have a very representative selection, which also coincides with other comparative works.

---

### Official Review · Reviewer_niK9 · 2022-10-25

**Confidence:** 4
**Correctness:** 4
**Technical Novelty And Significance:** 3
**Empirical Novelty And Significance:** 3
**Recommendation:** 6

**Clarity, Quality, Novelty And Reproducibility:**

Clarity is excellent, quality is good, novelty is good, reproducibility is good because the authors promise to release the code after review.

**Strength And Weaknesses:**

Strength:
1.	This paper is well motivated, aiming to tackle the problem of the unrealistic assumption of traditional disentangled representation learning.
2.	This paper proposes a new and effective training criterion through a Hausdorff distance term, which can be easily combined with existing disentanglement methods.
3.	The experiments show that the proposed method can improve the disentanglement and the out-of-distribution generalization ability.
Weaknesses:
1.	Some intuitions can be further explained, e.g., in section 2.2 the situation that breaks the factorized distribution can have a factorized support. It will be more convincing to give an example which does not have a factorized support will fail to disentangle, more intuitively show the relationship between factorized support and disentanglement.
2.	Since this paper claims to aim at the realistic scenario of disentangled representation learning, it is better to conduct experiments on real world datasets instead of the synthetic datasets(at least for the out-of-distribution setting.).
3.	The compared disentangled baselines seem to be out-of-date, it is better to incorporate the more recent disentangling methods.


**Summary Of The Paper:**

To avoid the unrealistic assumption of factor independence in traditional disentangled representation learning, this paper proposes to relax this assumption to factorized support, and proposes a Hausdorff-distance-based regularization. The authors conduct experiments on 3 classical datasets to show the improved disentanglement brought by the proposed method. Additionally, ablation studies show that the proposed method has  promising potential on out-of-distribution settings.

**Summary Of The Review:**

I have carefully checked the main body of this paper(including the introduction, method and experiments part.) I do not check the supplementary files.

---

> ### Author Response · Authors · 2022-11-09
> **RE: Official Review of Paper4621 by Reviewer niK9**
>
> We would firstly like to thank the reviewer for the detailed review and feedback, and address each noted question separately.
>
> ___Some intuitions can be further explained, e.g., in section 2.2 the situation that breaks the factorized distribution can have a factorized support. It will be more convincing to give an example which does not have a factorized support will fail to disentangle, more intuitively show the relationship between factorized support and disentanglement.___
>
> To provide a simple example:
>
> Suppose 2 ground truth factors are x,y coordinates of a feature in an image (e.g. coordinates of the center of the tip of the nose in the image of a person), where we suppose x in [a,b] and y in [a’,b’] holds over the entire distribution, with all combinations possible. The support of (x,y) is then the shape of an axis-aligned rectangle. Then latents z1=x, z2=x+y would be entangled, and their support is a parallelogram, which does not factorize. By contrast e.g. z1=tanh(x), z2=3.2 y  would be disentangled, and their support another axis-aligned rectangle, which factorizes.
>
> We hope this better qualitatively clarifies the connection between support factorization and disentanglement (please let us know if that is not the case!), and will work this example into the main paper.
>
> ___Since this paper claims to aim at the realistic scenario of disentangled representation learning, it is better to conduct experiments on real world datasets instead of the synthetic datasets(at least for the out-of-distribution setting.).___
>
> By “realistic”, we primarily refer to datasets that exhibit correlations. As this paper serves as a proof of concept, we require controlled scenarios, for which we can also model controlled distribution shifts (i.e. controlled out-of-distribution shifts).
> This is very hard to do when working with “real world” datasets. As such, we have to rely on benchmark datasets that give us control over the generative factors, which limits the selection of possible datasets to use to the ones utilized in our experimental study.
> But we agree that an important next step would be the transfer our insights to data collected from real generative processes, which we leave as a very interesting and relevant direction for future work to build on.
>
> ___The compared disentangled baselines seem to be out-of-date, it is better to incorporate the more recent disentangling methods.___
>
> We follow the disentanglement baselines used in other (recent) comparative works such as Locatello et al. (2019, 2021), Dittadi et al. (2021) or Traeuble et al. (2021) for which the selected baseline methods generally worked based.
> In addition, we are not aiming for overall state-of-the-art performance, but rather to show that we get (1) consistent relative improvements, (2) these improvements easily match and outperform other more involved regularizations, and (3) that HFS can be used as a regularizer also on top of these methods, which is indeed reflected in our experimental results.

---

### Official Review · Reviewer_rYAe · 2022-10-26

**Confidence:** 4
**Correctness:** 3
**Technical Novelty And Significance:** 4
**Empirical Novelty And Significance:** 3
**Recommendation:** 8

**Clarity, Quality, Novelty And Reproducibility:**

Clarity - The paper is clearly written and easy to read. Most of the decisions made are explained with justifications. Some of my concerns are mentioned in weaknesses above.

Quality - The research in the paper is of high quality. The paper describes a problem in previous works, proposes a reasonable solution and shows with extensive experimentation that it works.

Novelty - The proposed approach is the first to my knowledge to relax the independence assumption in previous disentangled representation learning works in a general manner (without explicit auxiliary variables or specific prior models).

Reproducibility - The authors provide a lot of their experimental details in the main paper and appendices. They also promise to release their code after publication.

**Strength And Weaknesses:**

Strengths
- The proposed relaxed assumption of factorization of support instead of independence provides more flexibility, allowing correlations in the training data to be learnt.
- The paper contains extensive experiments to support the proposed approach. Consistent improvements on Shapes3D, DSprites, MPI3D show the robustness of the proposed approach.
- Improvements in classification performance in “distributional shift” settings show that the proposed approach can facilitate generalization, especially in the case of strong correlation in the training distribution.
- The paper is well-written and easy to comprehend.


Weaknesses:

- The paper describes a problem in previous works, proposes a solution and shows that it works. This usually covers the most important components to evaluate an approach but the paper could be much more useful for the research community if the paper can provide a higher-level picture of where the proposed approach fits in this research area, for example:
    - A discussion on what different ways could provide the factorization of latents’ support and why the proposed HFS criterion is a better choice among them.
    - What are the drawbacks of the proposed approach? Should everyone replace the methods from previous works with the proposed method in all cases? If not, in what settings does the proposed approach fare better than others and in what settings it might be better to not use it? Does the proposed approach take more computational resources? Is it a more difficult objective to train?
- The paper draws its motivation from the fact that real-world datasets contain correlations between generative factors and therefore, the independence assumption is not very suitable for these real-world settings. But, the paper doesn’t include any experiments on uncontrolled real-world datasets, for example, a good first step could be the CelebA dataset.
- The paper mentions that previous “successful” works enforce the independence assumption even more strongly when it doesn’t hold for realistic data distributions. That should result in these approaches failing poorly on realistic datasets. Can the authors provide any findings (even from other papers) to support this argument? This again emphasizes the importance of at least one experiment with realistic data for this paper (where ideally, the previous works fail poorly, perhaps even more than vanilla VAE and the proposed approach works much better).
- The paper only considers one disentanglement evaluation metric — DCI-D. Even according to the cited justification Dittadi et al. (2021), both DCI and MIG are considered to be the best. Why do the authors choose to focus only on DCI? What are the results in terms of MIG? Also, it would be better if the evaluation metric(s) being used in the paper is explained briefly.
- Can the authors explain why HFS is working the best for no correlation setting? Shouldn't independence assumption work perfectly in this case and therefore, existing works have an advantage?
- Similarly, why do HFS versions of beta-VAE and beta-TCVAE work better than HFS? If the assumptions in beta-VAE and beta-TCVAE are not suitable for correlated settings, their criterion should hurt and not facilitate disentanglement, right?
- In Figure 2, I would have expected a consistent trend from top to bottom in the first few columns but there seem to be some training correlations in between (3rd, 7th, and 8th rows) that are better than the ones surrounding them. Why is that so? How exactly is the correlation made more severe in this case?
- Minor:
    - Definition 1.1 seems insufficient. This definition would be satisfied by the case when there is only one dimension that contains all the generative factors, the rest all being just noise. I think the authors meant one-to-one mapping between the latent dimensions and the generative factors (as mentioned in the next sentence). If so, this definition should be fixed accordingly.
    - It would be easier for the reader to understand if the authors can explain in more detail why the independence assumption doesn’t hold for the case in Figure 1, perhaps contrasting it with how the distribution would be in the case of independence.
    - Table 1, column 2, HFS is second best (bold) instead of beta-TCVAE+HFS.

**Summary Of The Paper:**

The paper proposes to relax the assumption of statistical independence used in many disentangled representation learning works. The paper proposes to only assume that the support of the latent factors’ distribution factorizes, which is a weaker constraint than statistical independence. Specifically, the paper proposes a training criterion that minimizes a Hausdorff set-distance between the latents’ actual support and its factorization. The paper contains extensive experiments on three datasets with varying degrees of correlation settings between ground truth generative factors. The proposed approach achieves better DCI-D scores as compared to previous works. The learnt representations from the proposed approach also achieve better classification performance in “distributional shift” settings.

**Summary Of The Review:**

I think the paper tackles an important problem in disentangled representation learning research, provides a reasonable solution and supports their approach with solid experiments. I have some concerns about the reasoning behind some experimental results, the lack of any experiments with realistic data and providing a more balanced picture of the proposed approach that I have described in more detail in the weakness section. But overall, I think the paper would be useful for the research community and hence I am inclining towards acceptance.

-------------------------------------------------------------------------------------------------------------------------------------------------------
The authors' response satisfactorily answers all my concerns. Therefore, I am keeping my acceptance recommendation.

---

> ### Author Response · Authors · 2022-11-09
> **RE: Official Review of Paper4621 by Reviewer rYAe - 1/4**
>
> We would firstly like to thank the reviewer for the detailed review and feedback, and address each noted question separately.
>
> ___The paper describes a problem in previous works, proposes a solution and shows that it works. This usually covers the most important components to evaluate an approach but the paper could be much more useful for the research community if the paper can provide a higher-level picture of where the proposed approach fits in this research area, for example:___
>
> ___(a) A discussion on what different ways could provide the factorization of latents’ support and why the proposed HFS criterion is a better choice among them.___
>
> ___(b) What are the drawbacks of the proposed approach? Should everyone replace the methods from previous works with the proposed method in all cases? If not, in what settings does the proposed approach fare better than others and in what settings it might be better to not use it? Does the proposed approach take more computational resources? Is it a more difficult objective to train?___
>
> __Regarding (a):__
> While one can perhaps specifically tackle the underlying family of networks to enforce support factorization, factorizing the support without having to account for architecture-specifics by solving for Eq. 2 naturally becomes a set-matching problem, for which minimization of set-distances is the straightforward solution. In this space, we agree that looking into other potential, non-Hausdorff set-based distances would have been interesting, but also would have added little additional novel information on the proof-of-concept that support factorization benefits disentanglement and consequently downstream OOD generalization.
> We also want to highlight that we did investigate a lot of other variants on the Hausdorff matching objective beyond our pair-based approximation in the supplementary, such as a soft matching objective, full support factorization with subsampling, probabilistic hausdorff variants and hausdorff variations using averaging to account for outliers (see. Appendix Tab. 2 and Fig. 5), where in all cases, our pairwise approximation proved best.
>
> __Regarding (b):__
> Indeed, there certainly is no free lunch, and this of course also holds for HFS:
>
> _Does the proposed approach take more computational resources?_
> The computation of our HFS objective does incur some additional computational overhead as noted in App. B, p.17 - while one training epoch using the beta-VAE objective only takes 52s, the use of HFS increases this time to 60s, which however still compares very favourably to other methods such as beta-TCVAE (70s) or FactorVAE (96s).
>
> _Is it more difficult to train?_
> We found little change in the overall training dynamics, with HFS actually being much more robust to hyperparameter value changes than the beta-value in the beta-VAE objective (see App. D, Fig. 6), and across all experiments and benchmarks, we consistently found HFS to be easiest to tune for when compared to beta-VAE, beta-TCVAE or FactorVAE, where the latter even trains a secondary adversarial network.
>
> _In what settings does the proposed approach fare better, and in which should one not use it?_
> In theory, if one does not expect any correlations in the training and test data to occur, and if the assumption of independent factors is accurate, then existing disentanglement methods which are based on this assumption, should fare better, whereas in settings with correlations, the (additional) use of HFS should help. However, in practice, we found both consistent (and high) improvements on correlated data, but also when applied to uncorrelated data, which indicates that the factorization of the support is a beneficial property to specifically account for even when the independence assumption holds. And as shown in the full tables in App. Tab. 5, we evaluated a lot of different, non-cherry-picked correlation settings, across the majority of which explicit support factorization helped. As such, based on our experimental results on controlled benchmark datasets, there is at the very least no major harm done in applying HFS beyond slight computational overhead, with a notable chance to improve disentanglement performance.

---

> > ### Author Response · Authors · 2022-11-09
> > **RE: Official Review of Paper4621 by Reviewer rYAe - 2/4**
> >
> > ___The paper draws its motivation from the fact that real-world datasets contain correlations between generative factors and therefore, the independence assumption is not very suitable for these real-world settings. But, the paper doesn’t include any experiments on uncontrolled real-world datasets, for example, a good first step could be the CelebA dataset.___
> >
> > As this paper aims to serve as a proof-of-concept, we required experiments with controlled correlations, for which we could vary both intensity and number of correlations. Consequently, we had to rely on benchmarks with full access to the generative factors (as also done e.g. in Traeuble et al. 2021), and as such, chose to forgo datasets such as CelebA, which were mined without an explicit license attached or permission of the persons photographed. We do agree that an important next step would be the transfer to data collected from real generative processes, which we leave as a very interesting and relevant direction for future work to build on.
> >
> >
> > ___The paper mentions that previous “successful” works enforce the independence assumption even more strongly when it doesn’t hold for realistic data distributions. That should result in these approaches failing poorly on realistic datasets. Can the authors provide any findings (even from other papers) to support this argument? This again emphasizes the importance of at least one experiment with realistic data for this paper (where ideally, the previous works fail poorly, perhaps even more than vanilla VAE and the proposed approach works much better).___
> >
> > Generally, we use the wording “realistic” to point to the existence of correlations in the data, which we simulate over a multitude of different correlation settings (Tab.2, App. Tab. 5), for which __we find a consistent drop in disentanglement performances across benchmarks as correlations increase__, which provides strong indication that correlation in the data is indeed a strong driver for reduced disentanglement performance. A similar behaviour is also seen in Traeuble et al. (2021), but to a lesser degree, as the investigated degrees and counts of correlation pairs are much less than what we investigated. In general, to the best of our knowledge, we are the first to perform such large-scale studies on the impact of changes in correlations and correlations shifts on downstream disentanglement (and generalization) performance.
> > And across all these settings, we then find consistent and in parts very high improvements in disentanglement performance through the explicit emphasis on support factorization.
> >
> > ___The paper only considers one disentanglement evaluation metric — DCI-D. Even according to the cited justification Dittadi et al. (2021), both DCI and MIG are considered to be the best. Why do the authors choose to focus only on DCI? What are the results in terms of MIG? Also, it would be better if the evaluation metric(s) being used in the paper is explained briefly.___
> >
> > The reviewer rightly points out that there are various disentanglement metrics that can be utilized, the majority of which however are generally strongly correlated (Locatello et al. 2019b, and as noted in the Related Works section), which preliminary experiments also verified, showing that the improvements in disentanglement measured on DCI-D also transfer to other disentanglement metrics such as MIG.
> > And as Locatello et al 2020a describes DCI-D as the most suitable disentanglement metrics, as well as other comparative works (Locatello et al 2019, 2020b, Traeuble et al. 2021, Dittadi et al. 2021) using DCI-D as the leading disentanglement metric, we thus chose to do the same, with thus little impact on the interpretation of the produced results.
> >
> > Finally, we do agree that a more detailed explanation of both the DCI-D disentanglement metric as well as other commonly utilized metrics is useful, and have included a summary in the supplementary materials section G.

---

> > > ### Author Response · Authors · 2022-11-09
> > > **RE: Official Review of Paper4621 by Reviewer rYAe - 3/4**
> > >
> > > ___Can the authors explain why HFS is working the best for no correlation setting? Shouldn't independence assumption work perfectly in this case and therefore, existing works have an advantage?___
> > >
> > > The factorization of the support is a general disentanglement property that holds regardless of whether we encounter correlation on the training data or not.
> > > As such, the fact that the disentanglement performance of HFS is best on the uncorrelated training data is to be expected, as there is no correlation shift to the uncorrelated test data.
> > > This is different to other models trained on correlated data, which experience increasing shifts in correlation (see e.g. Tab. 1 caption). This is also seen e.g. in Fig. 2, which shows that it is primarily the correlation shift that drives significant changes in test disentanglement.
> > >
> > > Separate to that, the reviewer is right to point out that the independence assumption should work best in this case, but on the other hand, there is also no explicit reasoning why a factorization of the support shouldn’t work.
> > > Though indeed, one would expect a slight drop in disentanglement performance on the uncorrelated data, instead of consistent improvements compared to the other methods.
> > > We attribute this to the hypothesis that HFS on its own is already sufficient to provide a reasonable degree of disentanglement, similar to other methods, but with less impact and constraint on the overall training dynamics of the VAE model, as also seen in the weighting robustness in Supp. D, Fig.6 - when the independence constraint is tuned up too high in standard disentanglement objectives, the reconstruction capabilities are severely impacted, much quicker resulting in broken training convergence.
> > >
> > > In addition, we find that when we use HFS as a regularizer on top of these independence-assuming approaches, we get even higher disentanglement performance, which we can attribute to the fact that while we do have correlations in most of our settings, we generally have various other factor pairs for which the independence assumption remains valid, and for which allowing for a small degree of factor independence benefits overall disentanglement performance, which supports HFS being a more relaxed constraint, and factorized support a generally beneficial property to account for.
> > >
> > >
> > > ___Similarly, why do HFS versions of beta-VAE and beta-TCVAE work better than HFS? If the assumptions in beta-VAE and beta-TCVAE are not suitable for correlated settings, their criterion should hurt and not facilitate disentanglement, right?___
> > >
> > > _(Info: This reply partly repeats from the previous question)_ We find that when we use HFS as a regularizer on top of these independence-assuming approaches, we get even higher disentanglement performance, which we attribute to the fact that while we do have correlations in most of our settings, we generally have various other factor pairs for which the independence assumption remains valid (as only some pairs of factors are correlated), and for which allowing for a small degree of factor independence benefits overall disentanglement performance, which supports HFS being a more relaxed constraint, and factorized support a generally beneficial property to account for.
> > > Conversely, as our benchmarks always have some factor pairs that remain (mostly) independent, the independence assumption does not generally become a breaking property, and can work well in conjunction with HFS.
> > > Finally, we do find among the highest __relative__ improvements in the 3-correlated-pairs and the shared confounder setting, i.e. where every factor exhibits a correlation to a shared confounding factor, meaning that there exist implicit correlations between all factors, and where the independence assumption holds least.  See e.g. 38% > 63.5% on Shapes3D with Shared Confounder, or 11.8% > 16.1% on MPI3D, among others.

---

> > > > ### Author Response · Authors · 2022-11-09
> > > > **RE: Official Review of Paper4621 by Reviewer rYAe - 4/4**
> > > >
> > > > ___In Figure 2, I would have expected a consistent trend from top to bottom in the first few columns but there seem to be some training correlations in between (3rd, 7th, and 8th rows) that are better than the ones surrounding them. Why is that so? How exactly is the correlation made more severe in this case?___
> > > >
> > > > That is a very valid point, and easiest explained by pointing to the detailed version of Fig. 2 in the supplementary, Fig. 10, which shows that the two main culprit rows, 7 and 8, belong to pairwise correlations with two pairs, but with a much smaller degree of overall correlation (sigma = 0.4 compared to sigma = 0.1). Our ranking on “correlation difficulties” was initially based on ranking by the number of correlated pairs first before accounting for the correlation strength sigma.
> > > > In retrospect, two slightly correlated pairs are likely the easier task compared to one strongly correlated pair, and these two rows can perhaps be moved further up the ordering, which would actually make the drop in performance and the disproportionate improvement in OOD classification performance visually even more evident.
> > > > For the 3rd row, this is likely just a statistical shift in performance due to the different choice of correlated ground truth factors used for that specific row, for which the model worked slightly better.
> > > >
> > > > ___(minor) Definition 1.1 seems insufficient. This definition would be satisfied by the case when there is only one dimension that contains all the generative factors, the rest all being just noise. I think the authors meant one-to-one mapping between the latent dimensions and the generative factors (as mentioned in the next sentence). If so, this definition should be fixed accordingly.___
> > > >
> > > > We agree that the Definition is in that sense somewhat ambiguous, and will update accordingly based on the suggestion. Thanks!
> > > >
> > > > ___(minor) It would be easier for the reader to understand if the authors can explain in more detail why the independence assumption doesn’t hold for the case in Figure 1, perhaps contrasting it with how the distribution would be in the case of independence.___
> > > >
> > > > Agreed. We have updated the associated section in the introduction to better highlight how satisfied independence would correspondingly look like.
> > > >
> > > > ___(minor) Table 1, column 2, HFS is second best (bold) instead of beta-TCVAE+HFS.___
> > > >
> > > > Indeed! Thanks for noticing, and fixed in the revised version.

---

> > > > > ### Comment · Reviewer_rYAe · 2022-12-12
> > > > > **Response to the authors**
> > > > >
> > > > > I thank the authors for their detailed responses to my questions. All my concerns are satisfactorily answered, therefore, I am keeping my acceptance rating.

---

### Author Response · Authors · 2022-11-09
**Shared response to all reviewers**

We would like to first thank the reviewers for their work and insightful comments, and their appreciation of the clear motivation behind our method and the idea of a factorized support for disentangled representation learning.
In addition, we thank the reviewers for acknowledging the quality and clarity of writing, as well as the high quality and relevance of our experimental studies to empirically highlight the benefits and robustness of our proposed pairwise support factorization objective to encourage disentanglement, as well as the consequential impact on generalization performance under strong distribution shifts.
For easy readability we have highlighted larger changes in the revised version of both the main paper and the supplementary, in red.

We address each set of reviewer comments in their associated thread.
Furthermore we want to highlight that after submitting our paper, we found that we had missed an important reference to related, independent work by Yixin Wang, Michael I. Jordan, “Desiderata for Representation Learning: A Causal Perspective” [ https://arxiv.org/abs/2109.03795 ].
In their causally motivated and primarily theoretical work, the authors arrive at the same principle of factorized support from an interventional causal perspective that explicits how correlations between factors can arise. In contrast, our work arrives at it from the perspective of relaxing the unrealistic independence assumption, to be robust under correlated factors, agnostic to how such correlation may arise.
We consider it as a strong positive sign when the same principle is arrived at independently from two quite different angles, and have updated our paper to properly reference, highlight, and discuss this work that predates ours.
Beyond their __different premise, different motivation and theoretical v.s. empirical focus__, our work proposes a _further pairwise relaxation_ to hold even when there are many factors, yielding a concrete, computationally tractable and statistically efficient criterion that is much better applicable in practical settings (c.f. Tab. 2 App.).
We also carry out a first large-scale evaluation with controlled correlations of factors to support our approach, and also link the associated increase in disentanglement with better generalization under distribution shifts and more data-efficient adaptation.

---

### Decision · Program_Chairs · 2023-01-20

**Decision:**

Accept: poster

**Justification For Why Not Higher Score:**

No experiments on real datasets.

**Justification For Why Not Lower Score:**

One of the first papers to present a method for learning disentangled factors that does away with independence assumption.

**Metareview: Summary, Strengths And Weaknesses:**

The paper moves away from the independence assumption in disentangled representation learning and relaxes it to factorized support assumption. It regularizes the Hausdorff distance between the latents actual support and its factorization to uncover correlated factors. Experiments are presented on synthetic datasets to demonstrate the effectiveness of the method in ground truth factor learning and in classification under controlled distribution shifts. While some reviewers expressed concerns over the use of synthetic datasets in the experiments, all reviewers are overall positive about the paper and acknowledge the novelty of the approach presented in the paper along with its effectiveness on the standard benchmarks used in the disentanglement literature.



**Note From Pc:**

if the above contains the word "oral" or "spotlight" please see: "oral" presentation means -> notable-top-5% and "spotlight" means -> notable-top-25%. As stated in our emails, we are disassociating presentation type from AC recommendations